# The global loss of floristic uniqueness

Qiang Yang [1]✉, Patrick Weigelt[2,3], Trevor S. Fristoe [1], Zhijie Zhang [1], Holger Kreft [2,4], Anke Stein[1], Hanno Seebens[5], Wayne Dawson [6], Franz Essl[7], Christian König [8], Bernd Lenzner [7], Jan Pergl [9], Robin Pouteau[10], Petr Pyšek[9,11], Marten Winter [12], Aleksandr L. Ebel[13,14], Nicol Fuentes[15], Eduardo L. H. Giehl [16], John Kartesz[17], Pavel Krestov[18], Toomas Kukk[19], Misako Nishino[17], Andrey Kupriyanov[20], Jose Luis Villaseñor[21], Jan J. Wieringa [22], Abida Zeddam[23], Elena Zykova[14] & Mark van Kleunen [1,24]

Regional species assemblages have been shaped by colonization, speciation and extinction over millions of years. Humans have altered biogeography by introducing species to new ranges. However, an analysis of how strongly naturalized plant species (i.e. alien plants that have established self-sustaining populations) affect the taxonomic and phylogenetic uniqueness of regional floras globally is still missing. Here, we present such an analysis with data from native and naturalized alien floras in 658 regions around the world. We find strong taxonomic and phylogenetic floristic homogenization overall, and that the natural decline in floristic similarity with increasing geographic distance is weakened by naturalized species. Floristic homogenization increases with climatic similarity, which emphasizes the importance of climate matching in plant naturalization. Moreover, floristic homogenization is greater between regions with current or past administrative relationships, indicating that being part of the same country as well as historical colonial ties facilitate floristic exchange, most likely due to more intensive trade and transport between such regions. Our findings show that naturalization of alien plants threatens taxonomic and phylogenetic uniqueness of regional floras globally. Unless more effective biosecurity measures are implemented, it is likely that with ongoing globalization, even the most distant regions will lose their floristic uniqueness.

[1] Ecology, Department of Biology, University of Konstanz, Konstanz, Germany. [2] Biodiversity, Macroecology & Biogeography, University of Göttingen, Göttingen, Germany. [3] Campus-Institut Data Science, Göttingen, Germany. [4] Centre of Biodiversity and Sustainable Land Use, University of Goettingen, Göttingen, Germany. [5] Senckenberg Biodiversity and Climate Research Centre, Frankfurt, Germany. [6] Department of Biosciences, Durham University, Durham, UK. [7] Bioinvasions, Global Change, Macroecology Group, Department of Botany and Biodiversity Research, University of Vienna, Vienna, Austria. [8] Ecology and Macroecology group, University of Potsdam, Potsdam, Germany. [9] Czech Academy of Sciences, Institute of Botany, Department of Invasion Ecology, Průhonice, Czech Republic. [10] AMAP, Univ Montpellier, CIRAD, CNRS, INRAE, IRD, Montpellier, France. [11] Department of Ecology, Faculty of Science, Charles University, Prague, Czech Republic. [12] German Centre for Integrative Biodiversity Research (iDiv) Halle-Jena-Leipzig, Leipzig, Germany. [13] Department of Botany, Tomsk State University, Tomsk, Russia. [14] Central Siberian Botanical Garden, Siberian Branch of Russian Academy of Sciences, Novosibirsk, Russia. [15] Departamento de Botánica, Facultad de Ciencias Naturales y Oceanograficas, Universidad de Concepción, Concepción, Chile. [16] Departamento de Ecologia e Zoologia, Federal University of Santa Catarina, Florianópolis, Brazil. [17] Biota of North America Program, Chapel Hill, NC, USA. [18] Botanical Garden-Institute FEB RAS, Vladivostok, Russia. [19] Institute of Agricultural and Environmental Sciences, Estonian University of Life Sciences, Tartu, Estonia. [20] Institute of Human Ecology, Siberian Branch of Russian Academy of Sciences, Kemerovo, Russia. [21] Departamento de Botánica, Universidad Nacional Autónoma de México, Mexico City, Mexico. [22] Naturalis Biodiversity Centre, Leiden, The Netherlands. [23] Ingenieur en Ecologie vegetale, Algiers, Algeria. [24] Zhejiang Provincial Key Laboratory of Plant Evolutionary Ecology and Conservation, Taizhou University, Taizhou, China. ✉email: qiang.yang@uni-konstanz.de

The intentional or accidental introduction of organisms by humans[1–3] has enabled species to overcome natural biogeographic barriers[4]. The alien species that subsequently overcome environmental and reproductive barriers, and thus have established self-sustaining populations (i.e., have become naturalized[5]) alter the composition of species assemblages. These naturalized species can have ecological and evolutionary consequences in their new regions through e.g., changes in biotic interactions and hybridization[6,7]. Furthermore, they also change patterns of biotic uniqueness or distinctiveness of those regions compared to others. This means that the natural borders between biogeographic realms may move or disappear. The loss of a region's biotic distinctiveness could also have economic consequences, as it might make a region less attractive for tourists[8]. Human-caused biotic homogenization (i.e., increased similarity in species composition between regions) across large spatial scales has been documented in a number of animal groups and for ecological networks[1,9–11]. However, for vascular plants, no study on floristic homogenization at the global scale has been completed (but see refs. [12,13] for continental-scale studies).

A naturalized species can change a region's floristic uniqueness in several ways (Fig. 1). First, it can increase floristic similarity of two regions (i.e., resulting in floristic homogenization) when the species is native to one of the two regions and naturalizes in the other, or when it is not native to both regions and naturalizes in both. Second, it can decrease the floristic similarity of two regions (i.e., resulting in floristic differentiation) when the species is not native to both regions but naturalized in only one of them. The net change in floristic similarity will thus depend on the sizes of the different sets of naturalized species in both regions. Floristic similarities of regions could in principle also change due to regional extinctions of native species. However, while many native plant species have been extirpated from local communities, the numbers of species that have entirely disappeared from regional floras are usually an order of magnitude lower than the number of naturalized alien species[12]. As a consequence, patterns in the degree of floristic homogenization (or differentiation) of regions are primarily driven by naturalized alien species[12].

The uniqueness of a regional flora is not only characterized by the numbers of species that it does and does not share with other regions (i.e., taxonomic uniqueness). It is also characterized by the distinctiveness of the evolutionary history captured by its species (i.e., phylogenetic uniqueness). In other words, a region's flora is even more unique when a species is endemic to that region and there are no close relatives in other

regions. This means that naturalization of a species closely related to some of the native species (for example, a congeneric species) will impact the phylogenetic floristic uniqueness of the region to a lesser extent than naturalization by a distantly related species (Supplementary Fig. 1). Taxonomic uniqueness, however, will be affected to the same degree, irrespective of the phylogenetic distance between the naturalized and native species (Supplementary Fig. 1)[12,14].

The degree and direction of change in floristic similarity between two regions likely depends on both biogeographic (e.g., climatic similarity, geographic distance) and anthropogenic factors (e.g., exchange of goods and people). It is well established that natural floristic similarity exponentially decreases with geographic distance, as nearby regions share more species than isolated, distant ones[15–17]. As human introductions of alien species help them to overcome natural barriers to dispersal, naturalizations are likely to weaken the relationship between floristic similarity and geographic distance. Furthermore, as climatic suitability has consistently been shown to be a major determinant of the establishment likelihood of alien species[18], it is likely that stronger homogenization will be found between regions with more similar climates due to preadaptation of the alien species. This should be particularly the case for climatic counterparts that are geographically distant and thus share few native species. Finally, globalization, and the associated increases in human trade and travel, is a major driver of the introduction of alien organisms[19–21]. Therefore, regions with greater exchange of goods and people are more likely to have increased floristic similarities. This is likely reflected in the importance of current and past administrative relationships between regions, where past relationships mainly reflect the previous colonial European empires. For example, the British global empire had a network of 126 botanical gardens that exchanged plant species[22]. Moreover, belonging to the same colonial empire roughly doubled the trade flow between regions[23].

Here, we used data on native and naturalized alien floras of 658 regions (e.g., countries, states, provinces) around the world to quantify changes in taxonomic and phylogenetic similarities between regional floras caused by naturalized alien plants. We then analyzed how these changes in floristic similarity are associated with geographic distance, climatic distance, and administrative relationships between regions. Furthermore, we analyzed how the mean extent of homogenization of a region relates to its size, native species richness, degree of endemism, naturalized species richness, and insularity. We show that the naturalization of alien plants reduces taxonomic and phylogenetic similarities between regional floras, causing floristic homogenization globally. The degree of floristic homogenization between two regions increases with their geographical distance, climatic similarity, and past and current administrative relationships.

## Results

**Changes in floristic similarity.** The 658 regions for which we had reliable data on both native and alien naturalized floras covered 110 countries (~65.7% of the ice-free land surface) and 189,762 flowering-plant species (~62.3% of all the flowering plant species in The Plant List;[24] Supplementary Fig. 2). For each pair of regions, we quantified the taxonomic and phylogenetic similarity between their native floras ($SimTax_{native}$, $SimPhyl_{native}$) and between their combined native and naturalized floras ($SimTax_{native+naturalized}$, $SimPhyl_{native+naturalized}$), as well as the change in similarity (i.e., degree of homogenization) caused by naturalization of alien species ($H$, calculated as log-response ratios of $SimTax_{native+naturalized}$ and $SimTax_{native}$, and of $SimPhyl_{native+naturalized}$ and $SimPhyl_{native}$). We found that alien

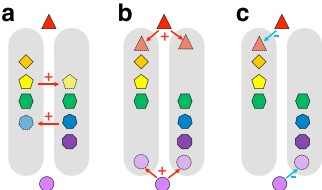

**Fig. 1 Illustration of different scenarios of species naturalization and their consequences on similarity between two regions.** In **a**–**c**, the triangle and circle represent species that are non-native to the two regions, denoted by the gray areas, while the other symbols represent native species of the regions. In scenario **a**, species that are native to one of the two regions have become naturalized aliens in the other region, resulting in homogenization. In scenario **b**, species not native to both regions have become naturalized in both, resulting in homogenization. In scenario **c**, each of the two non-native species has become naturalized in a different region, resulting in differentiation. The symbols "+" and "−" indicate whether the naturalized species increase similarity (homogenization) or reduce similarity (differentiation), respectively.

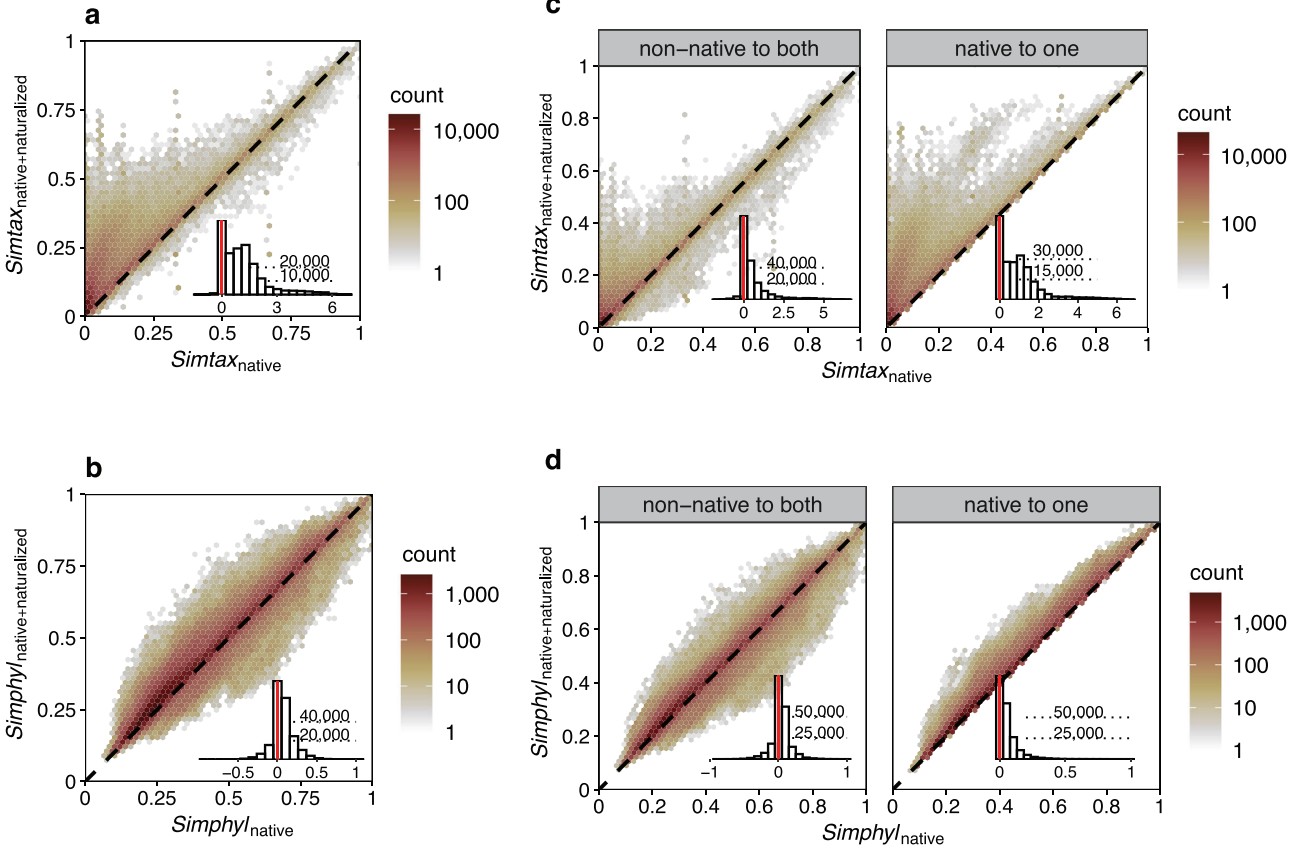

**Fig. 2 Changes in taxonomic and phylogenetic similarity of regional floras due to naturalized alien plants. a** and **b**, hexagonal bin plots showing changes in taxonomic and phylogenetic similarity, respectively, driven by all naturalized alien species. The dashed line indicates where the similarity of two native floras ($Sim_{native}$) equals the similarity of their native and naturalized floras combined ($Sim_{native+naturalized}$). The data ($n = 216{,}153$ region × region comparisons) are binned into hexagonal cells to improve figure readability. The inset histograms show the frequency distribution of the change in taxonomic and phylogenetic similarity, respectively. **c** and **d** also show the change in taxonomic and phylogenetic similarity, respectively, but for the subsets of naturalized species that are either restricted to species that are not native to both regions ('non-native to both', corresponding to scenarios **b** and **c** in Fig. 1) or to species that are native to only one of the two regions ('native to one', corresponding to scenario a in Fig. 1).

species have increased the taxonomic similarity in 90.7% ($n = 196{,}154$) of all region pairs ($n = 216{,}153$; Fig. 2a) and the phylogenetic similarity (i.e., phylogenetic homogenization) in 77.2% ($n = 166{,}789$) of all region pairs (Fig. 2b). The median changes in taxonomic and phylogenetic similarities due to naturalized alien species were 206% and 5.3%, respectively (Supplementary Fig. 3). The change in phylogenetic similarity is much smaller than the change in taxonomic similarity (paired Wilcoxon test: $n = 216{,}153$, $V = 2.05 \times 10^8$, $P < 0.001$), because phylogenetic similarities of native floras are higher than the corresponding taxonomic similarities for most region pairs (compare Fig. 3a and d; paired Wilcoxon test: $n = 216{,}153$, $V = 4.90 \times 10^5$, $P < 0.001$).

When considering only species that are native to one and naturalized in the other region (scenario a in Fig. 1, which can only result in an increase in similarity), the median changes in taxonomic and phylogenetic similarities are 175.5% and 3.1%, respectively (Fig. 2c, d). When considering only naturalized species that are not native to both regions (scenarios b and c in Fig. 1), we found that 153,415 (71.0%) and 151,885 (70.3%) of the region pairs increased, whereas 44,233 (20.5%) and 63,920 (29.6%) of the pairs decreased their taxonomic and phylogenetic similarities, respectively (Fig. 2c, d). For the scenarios of species that are not native to both regions, the median changes in taxonomic and phylogenetic similarities are 41.0% and 3.1%, respectively.

**Factors driving pairwise floristic similarities of regions and changes therein.** Taxonomic and phylogenetic similarities of native floras exponentially declined with geographic distance and with climatic distance (Fig. 3a, b, d, e and Supplementary Table 1). The same was true for the native and naturalized floras combined (Fig. 3a, b, d, e and Supplementary Table 1). However, the natural biogeographic pattern of geographic distance decay in similarity became less steep when the naturalized alien species were added (Fig. 3a, d and Supplementary Table 1). More specifically, alien species increased the halving distance (i.e., the distance at which similarity decreases by 50%[17]) of taxonomic similarity from 1791 to 2683 km and that of phylogenetic similarity from 7008 to 7489 km.

Next, we calculated an index of homogenization as the log-response ratio of floristic similarity of native and naturalized species combined vs floristic similarity of native species only. This confirmed that particularly geographically distant floras have become less distinct from each other due to naturalized plants (Fig. 3c, f, Supplementary Figs. 4, 5 and Supplementary Table 2). Furthermore, the more climatically distant the two regions are the smaller their degree of taxonomic and phylogenetic homogenization. This, however, was only true for pairs of geographically distant regions (Fig. 3c, f, Supplementary Figs. 4, 5, and Supplementary Table 2). The negative association between floristic homogenization and climatic distance was mainly due to differences in temperature and, to a lesser extent, due to

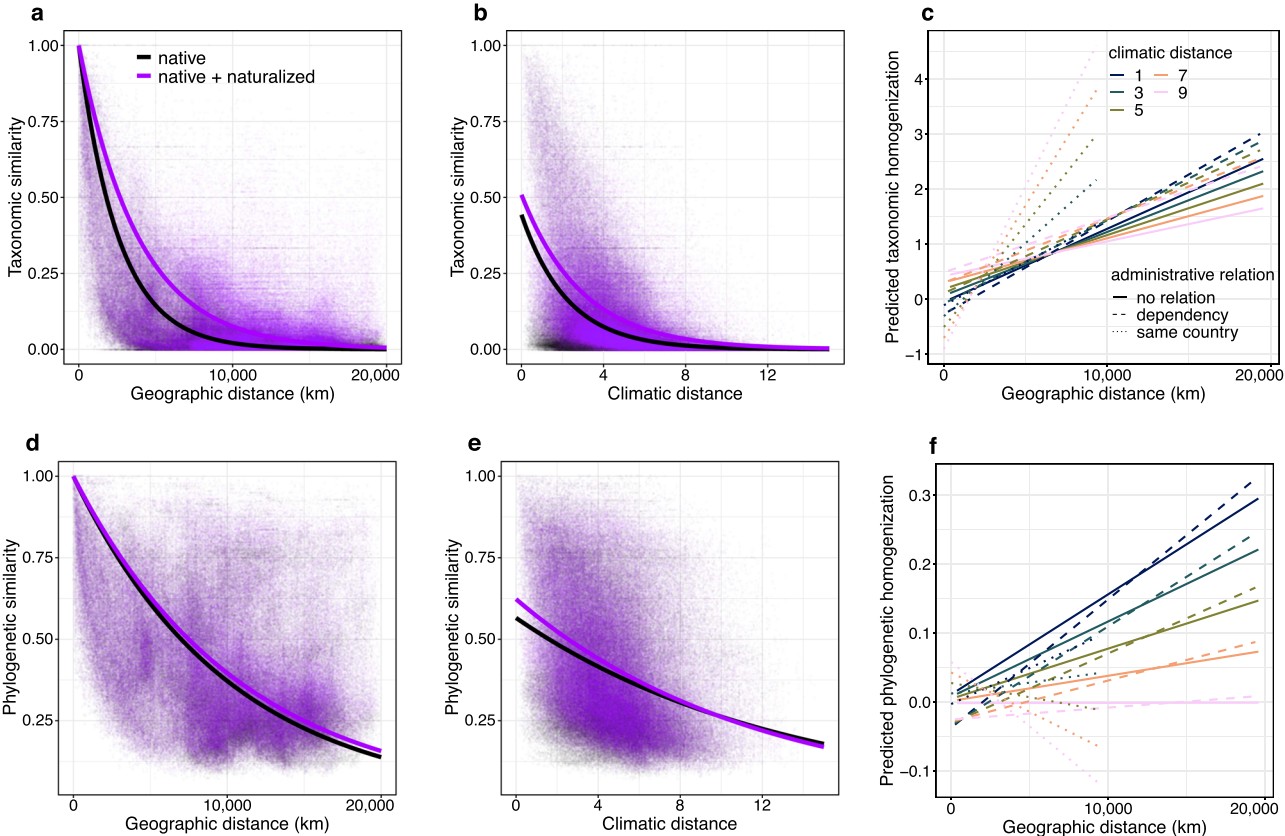

**Fig. 3 Floristic similarities and changes therein due to naturalized alien species in relation to geographic distance, climatic distance and administrative relationships.** Taxonomic (**a**, **b**) and phylogenetic (**d**, **e**) similarities of native species (black) and of natives and naturalized species combined (purple) versus geographic (**a**, **d**) and climatic (**b**, **e**) distance between the two regions in a pair ($n = 216,153$ for each type of similarity). The curves in **a**, **b**, **d**, and **e** are from the fitted GLMs (see "Methods"). Changes in taxonomic and phylogenetic similarity (i.e., the degree of homogenization) (**c**, **f**) as predicted by the multiple regression on distance matrices showing the effects of geographic distance, climatic distance, the administrative relationship, and their interactions. To better visualize the predicted response of homogenization to the predictors, we arbitrarily set climatic distance values to 1, 3, 5, 7, and 9, which correspond to the 2.2%, 25.0%, 60.9%, 87.0%, and 96.8% quantiles of the climatic distance values of all region pairs, respectively. Administrative relationships include pairs of regions (1) belonging to the same country, (2) that have a current or past dependency on the other, and (3) without administrative ties. Dependency refers to the relationship in which two regions are either currently dependent territories or past colonies of the same country. It also refers to two regions that belong to two different countries of which one is or was the dependent territory or colony of the other.

differences in precipitation (Supplementary Fig. 7c, d, g, h and Supplementary Fig. 8). While phylogenetic homogenization was also negatively associated with pairwise differences in precipitation, taxonomic homogenization was actually slightly positively associated with differences in precipitation (Supplementary Fig. 7d, h and Supplementary Fig. 8).

As a crude indicator of the exchange of goods and people between regions, we used the current and past administrative links between regions. The floras of regions with current or past administrative links have taxonomically become more similar to each other than the floras of regions with no such links. In particular, the degree of taxonomic homogenization increased more rapidly with geographic distance for region pairs belonging to the same country than for other region pairs (slope comparison using ANCOVA: $P < 0.001$; Fig. 3c, Supplementary Figs. 4a, 5d and Supplementary Table 2). Moreover, regions that are not part of the same country but have either past administrative relationships (e.g., regions that were part of the same colonial empire) or current ones (e.g., overseas territories), also showed a slightly stronger increase in the degree of taxonomic homogenization with geographic distance than pairs of regions without administrative relationships (slope comparison using ANCOVA: $P < 0.001$; Fig. 3c, Supplementary Figs. 4a, 5d, and Supplementary Table 2). For the degree of phylogenetic homogenization, on the

other hand, the association with administrative links between regions was less clear (Fig. 3f, Supplementary Figs. 4b, 5h, and Supplementary Table 2).

**Homogenization hotspots and their association with characteristics of the regions and floras.** We identified hotspots of taxonomic homogenization (i.e., regions with the greatest average increase in taxonomic similarity with other regions due to naturalized alien species) in Australasia (including regions in New Zealand, southwestern Australia, and southeastern Australia) and on many oceanic islands (Fig. 4a and Supplementary Fig. 9a). While hotspots of phylogenetic homogenization coincided with hotspots of taxonomic homogenization for mainland regions, this was not the case for islands (Fig. 4 and Supplementary Fig. 10). In particular, relatively small oceanic islands that are hotspots of taxonomic homogenization, showed significantly less phylogenetic homogenization than most mainland regions ($P < 0.001$, Supplementary Fig. 9b). Moreover, the few regions that showed phylogenetic differentiation are almost exclusively islands (Fig. 4b and Supplementary Fig. 9b).

The high mean taxonomic homogenization on islands might reflect that islands are usually small, have a low native species richness with a relatively high proportion of endemics, and have a

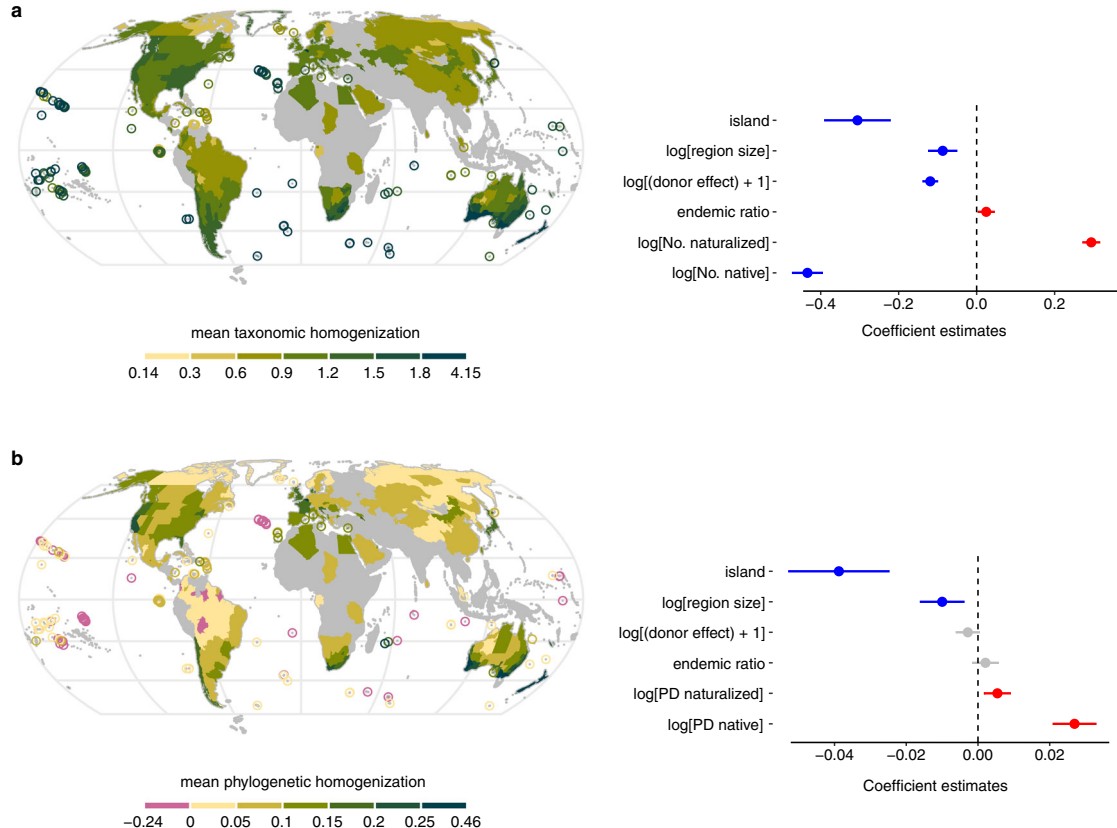

**Fig. 4 Mean changes in taxonomic and phylogenetic similarity (degree of homogenization) of regions around the world, and how these depend on characteristics of the regions and their floras.** For each of the regions ($n = 658$), we calculated the mean value of its pairwise change in floristic similarity with other regions. The color gradients indicate the level of the mean degree of homogenization (or differentiation, in the case of negative values). Island regions smaller than 10,000 km$^2$ ($n = 137$) are marked with circles to ensure their visibility on the map. The plot to the right side of each map shows the strength of the association between mean degrees of homogenization with the diversity of the native and naturalized floras (species richness values for the model on taxonomic homogenization, and phylogenetic diversity (PD) values for the model of phylogenetic homogenization), the proportion of endemic species, the donor score of the region (i.e., the average number of non-native regions in which each native species is naturalized in), the size of the region, and whether the region is an island or a mainland region. In each plot, points represent the coefficient estimates and the error bars represent the 95% confidence interval. The red, blue, and gray colors indicate significantly positive, significantly negative, and insignificant associations, respectively. The *P* values of the explanatory variables in (**a**) from the top: $6.49 \times 10^{-12}$, $7.29 \times 10^{-6}$, 0, 0.03, 0, and 0. The *P* values of the explanatory variables in (**b**) from the top: $1.09 \times 10^{-7}$, $1.84 \times 10^{-3}$, 0.10, 0.28, $5.03 \times 10^{-3}$, and 0. Note that the *P* value smaller than $2.2 \times 10^{-16}$ (the minimum *P* value that returned from the function *summary* in R) is written as 0 here.

relatively high naturalized species richness. Indeed, after controlling for those variables, the effect of being an island instead of a mainland region on taxonomic homogenization became negative (Fig. 4a). The mean taxonomic homogenization also decreased significantly with greater region size, native species richness, and the donor role of the region (Fig. 4a). Furthermore, taxonomic homogenization significantly increased with greater proportion of endemic native species and naturalized species richness (Fig. 4a). The mean phylogenetic homogenization of regions also decreased significantly with greater region size (Fig. 4b). Furthermore, it increased significantly with greater phylogenetic diversity of both the native species and the naturalized species (Fig. 4b).

## Discussion

Biological invasions and the naturalization of alien species have been suggested to result in a New Pangea (i.e., a world without major biogeographic barriers)[25]. We indeed found evidence of floristic homogenization for regional floras around the world and showed that pairwise floristic homogenization of regions is strongest for geographically distant regions with similar climates. Moreover, pairwise taxonomic homogenization was strongest for regions with current or historical administrative links than for

regions without such links. Hotspots of floristic homogenization were mainly found in Australasia, and while many oceanic islands were also hotspots of taxonomic homogenization, they were at the same time coldspots of phylogenetic homogenization. So, the loss of floristic uniqueness varies around the globe, and may differ for taxonomic and phylogenetic indices.

Although a naturalized alien species could theoretically contribute to floristic differentiation (Fig. 1 and Supplementary Fig. 1), we found that in more than 90% and 77% of the region pairs, naturalizations has increased taxonomic and phylogenetic similarities, respectively. It is likely that particularly species that are naturalized in many regions and are frequently considered to be invasive, and which usually also have large native ranges[26,27], contribute most to floristic homogenization. Indeed, removal of the 10% most widely naturalized species resulted in significant reductions in taxonomic and phylogenetic similarities, whereas removal of the 10% least widely naturalized species actually resulted in slightly higher taxonomic and phylogenetic similarities (Supplementary Fig. 11). In other words, as expected, global homogenization of regional floras is mainly driven by the widely naturalized species, whereas the rare ones tend to contribute to differentiation.

With increasing geographic distance, natural biogeographic barriers (e.g., oceans, mountain ranges) between regions are likely to add up to limit dispersal. As a consequence, we found —like many other studies[15–17]— that there is a natural exponential decline in floristic similarity with increasing geographic distance. As humans have enabled plants to circumvent biogeographic barriers, the natural decline in floristic similarity with distance is weakened by naturalized species. As distant regions share few native species, an increase in their number of shared species due to naturalizations has a stronger effect on floristic similarity than is the case for near regions.

Our results show that floristic homogenization was most pronounced for distant regions with similar climates (Fig. 3c, f). This emphasizes the important role of climate matching in alien plant naturalization[28]. An exception, however, was pairwise taxonomic homogenization for regions that are part of the same country. Here, the degree of homogenization increased with climatic distance. This counter-intuitive result may reflect that geographic distances between regions within countries are relatively small, and that nearby, climatically similar regions, are likely to have similar native floras. As a consequence, alien species will have relatively minor impacts on the already high floristic similarity of those regions.

Because of geographical barriers and long geological separation, distant regions have evolved distinct phylogenetic clades. The strong negative effect of climatic distance on phylogenetic homogenization could reflect that although different phylogenetic clades evolved in isolated regions, they adapted to similar climates (i.e., convergent evolution). So, they are preadapted to each other's native ranges, and introduction by humans gave these phylogenetically distinct species the opportunity to naturalize and to over-proportionally contribute to the phylogenetic similarity of these regions. For example, many of the *Eucalyptus* species, which are endemic to Australia, are now widely naturalized in climatically similar regions on other continents[29].

Our estimate of a single overall climatic distance was based on all available bioclimatic variables. When using instead the differences in scores along the first two axes of a principal component analysis (PCA) on the bioclimatic variables, we found that the effect of temperature differences (PC1) on floristic similarity was stronger than the effect of precipitation differences (PC2). Moreover, the effects of temperature differences on homogenization were very similar to the patterns for overall climatic distance. The effects of precipitation differences on phylogenetic homogenization were also very similar to the patterns for overall climatic distance. However, the effects of precipitation differences on taxonomic homogenization were opposite to the pattern for overall climatic similarity. In other words, taxonomic homogenization was higher between regions with different precipitation conditions. This effect, however, was relatively weak when compared to the opposite effect of temperature differences, particularly for geographically distant regions (Supplementary Fig. 7c, d). The apparently lower importance of precipitation, compared to temperature, might reflect that many naturalized aliens are weeds on irrigated agricultural lands and along rivers[30], where they may depend less on natural precipitation.

Colonialism, predominantly by Europeans, and trade are widely acknowledged to be responsible for the introduction and spread of alien organisms[19–22,31]. As past and current trade are usually more intense between regions that had or still have administrative links[23], we expected that these links would result in larger degrees of floristic homogenization. Indeed, the degree of taxonomic homogenization increased more rapidly with geographic distance for region pairs belonging to the same country than for other region pairs. This most likely reflects facilitated spread of an alien species within a country after it has become established in one of its regions. Moreover,

regions not under the same governance, but with either past or current administrative relationships, also showed a slightly stronger increase in the degree of taxonomic homogenization with geographic distance than regions without any administrative relationships. While there were no clear patterns for phylogenetic homogenization in this regard, our results overall reveal that patterns of global floristic homogenization contain signals of colonial history and current administrative relationships.

When we calculate the mean degrees of homogenization for each region, we found that several Australasian regions are hotspots of taxonomic homogenization. This most likely reflects that Australasia's long biogeographic isolation has resulted in a unique native flora[32]. As a consequence, the many alien species introduced there by the European settlers and government agencies[33,34] have had relatively strong effects on Australasia's floristic similarity to the rest of the world. Many oceanic islands, which due to extreme isolation usually have unique and species-poor floras with large proportions of endemics[35], also emerged as hotspots of taxonomic homogenization. This is in line with previous findings that many islands, and particularly distant ones, harbor disproportionally high numbers of naturalized alien plants[3,36].

Interestingly, while hotspots of phylogenetic homogenization coincide with hotspots of taxonomic homogenization for mainland regions, this was not the case for islands. Relatively small oceanic islands that are hotspots of taxonomic homogenization, showed significantly less phylogenetic homogenization than most mainland regions, and some of them even showed differentiation. This is mostly due to the generally high phylogenetic similarities that native floras on islands exhibit to other floras (Supplementary Figs. 12 and 13)[17]. Many of the small islands are oceanic islands, which emerged from the ocean by volcanic activities or by uplifting of the ocean floor due to tectonic plate movement. This means that native floras on oceanic islands are the result of natural colonization events, particularly from mainland areas, and subsequent speciation events[37]. Therefore, island floras are phylogenetic subsets of other floras that served as donors[37]. Moreover, as the chance of speciation on an island increases with island size and time[38], it is unlikely that many phylogenetically unique species that are distantly related to native floras in nearby regions will have evolved on small and relatively young islands.

While islands had higher mean degrees of taxonomic homogenization than mainland regions, this association reversed after accounting for other characteristics of the regions and their floras. This shows that the high taxonomic homogenization on islands is mainly due to the fact that islands are small, have relatively few native species with a large proportion of endemics[35], and have relatively high numbers of naturalized species[36]. We also found that regions with a strong role as donors of naturalized plants have undergone less taxonomic homogenization. This is surprising given that the species donated by those regions should not only contribute to the homogenization of the recipient regions but also to the homogenization of the donor region itself. Most likely, our finding reflects that regions in Australasia are taxonomic homogenization hotspots because they have received many alien species[36] but have donated fewer species to other parts of the world than expected based on the sizes of their floras[3]. This also highlights that there is an asymmetry in the donor and recipient roles of regions.

For the mean extent of phylogenetic homogenization of regions, we found overall similar associations with characteristics of the regions and floras as we found for mean taxonomic homogenization. The associations with donor role and proportion of endemic species, however, were not statistically significant, and the phylogenetic diversity of the native flora was positively associated with the mean phylogenetic homogenization. This

might reflect that in the calculation of the pairwise phylogenetic Simpson similarity index only the lowest phylogenetic diversity of the two regions is used in the denominator (see Eq. 2 in "Methods"). For regions with a large native phylogenetic diversity, this means that their pairwise phylogenetic similarity to other regions is determined by the phylogenetic diversity of those other—smaller—regions. As a consequence, the change in phylogenetic similarity due to naturalized species will be relatively large for regions with a high phylogenetic diversity.

While our analysis is global in extent, there is a lack of high-quality data on native and naturalized alien species lists for some regions, particularly in Africa, Eastern Europe, and tropical Asia. Consequently, 37.7% of species in the extant global flora were not included in our analysis (Supplementary Fig. 2) and 34.3% of the ice-free terrestrial surface was not covered by our regions (Fig. 4). We cannot exclude the possibility that this might have caused some biases in the pattern of homogenization. For example, the relatively low number of regions with data in tropical Africa and South-East Asia prevents from many potential comparisons of tropical South American regions with other geographically distant tropical regions. As a consequence, the degree of homogenization for tropical South American regions might be underestimated. Nevertheless, as the overall representativeness of our data is high, these potential biases are unlikely to change the overall finding of a loss of regional floristic uniqueness. Another limitation is that species lists are mainly available for administrative regions that vary markedly in size, environmental heterogeneity and native species richness. The same would also be true for biogeographic regions, but for the latter one would expect a higher dissimilarity in their native floras. The effect of homogenization by naturalized species would hence be potentially bigger when looking at bioregions. Theoretically, however, changes in species' occurrences throughout a region or their local abundances could also influence regional floristic homogenization. If one could account for abundance, the naturalized species that have become widespread and invasive (i.e., abundant) would contribute more strongly to floristic homogenization. It should be noted, though, that McKinney and Lockwood[39] found that taxonomic similarity values based on abundance data were strongly correlated with those relying on presence–absence data. So, the results for regional floristic homogenization might not be strongly affected by variation in the abundance of the naturalized species. However, we also note that the degree of homogenization of local communities within and across regions may vary. The degree to which this is the case will depend on how widespread the aliens, as well as the natives, are within the regions and on the resulting patterns of species co-occurrence.

While it has frequently been suggested that human-caused biotic exchange has resulted in a New Pangea or Homogecene[6,40], a quantitative analysis of the degree of floristic homogenization was still missing. We here show that plant naturalizations have resulted in taxonomic and phylogenetic homogenization of floras worldwide. We found that the degree of floristic homogenization between two regions increases with their geographical distance, climatic similarity, and past and current administrative relationships. Furthermore, we showed that global homogenization hotspots are mostly regions with unique native floras (e.g., regions of Australasia and many oceanic islands). Introduction and naturalization of alien species are ongoing processes and likely to increase steadily for decades to come[41]. Therefore, floristic homogenization is likely to continue and even accelerate[42,43], with largely unknown ecological, evolutionary and socioeconomic consequences. Instead of waiting for the 'Homogecene'[40], action is needed to preserve the biotic uniqueness of regions globally[6,44]. This will require stronger biosecurity regulations for trade and transport and protection of native vegetation.

## Methods

**Quantification of changes in floristic similarity.** To quantify changes in floristic similarity by naturalized flowering plant species, we extracted regional lists of alien species from the Global Naturalized Alien Flora (GloNAF) database[45] and regional lists of native species from the Global Inventory of Floras and Traits (GIFT) database[46]. The GloNAF database contains lists of naturalized vascular plant taxa for 861 regions (countries or subnational administrative units), ranging in size from 0.03 to 6,864,961 km$^2$ (median size is 15,152 km$^2$) and covering >80% of the terrestrial ice-free surface globally[47]. GloNAF includes 13,803 plant taxa that, according to the original data sources, are alien plants and have established self-sustaining wild populations in the respective regions (i.e., are naturalized[5]). The GIFT database is a compilation of floras and checklists of predominantly native vascular plant species with an indication of their floristic status for more than 300,000 species across nearly 3000 regions with near global coverage[46]. We first selected regions that matched perfectly between GloNAF and GIFT. Additionally, we merged some GloNAF regions to match a larger GIFT region, and vice versa, by comparing the overlapping area of nested regions using the R package 'sf' (version 0.8-0)[48].

To ensure the highest data quality, and to be on the conservative side, we restricted our analysis to regions with complete or nearly complete checklists of both native and naturalized alien species. For GloNAF, we only included regions for which there was at least one species list judged to include more than 50% of the naturalized taxa for that region[45]. Although the judgment of species-list completeness is coarse and for most lists made by the GloNAF curators, it allows the exclusion of regions for which the data are obviously poor. For GIFT, we included a region only if at least one species list aimed to represent its entire native angiosperm flora. Our strict selection criteria resulted in a dataset including native and naturalized species for 658 non-overlapping regions, including 154 island regions, 503 mainland regions and one region including both islands and mainland areas (Chile). These regions covered all continents, except Antarctica, but there was low coverage for parts of Africa and Asia (Fig. 4).

We restricted our analyses to flowering plants (angiosperms), which had the most complete species lists, and to species with accepted names in The Plant List[24] (http://www.theplantlist.org/). We excluded species with an uncertain native/alien status or with a conflicting status, i.e., being native to a region according to GIFT but being alien to the same region according to GloNAF. Furthermore, since the native/alien status of many infraspecific taxa and hybrid taxa are less clear, we restricted our analyses to the species level (i.e., infraspecific taxa were assigned to the binomial species name), and we excluded hybrids. Our final dataset included 1,139,254 native species-by-region records for 189,110 species and 141,762 naturalized species-by-region records for 10,130 species.

For all 216,153 possible pairwise combinations of the 658 regions, we quantified the taxonomic and phylogenetic similarities between their native floras ($SimTax_{native}$, $SimPhyl_{native}$), and between their floras including both native and naturalized alien species ($SimTax_{native+naturalized}$, $SimPhyl_{native+naturalized}$). As the regions vary largely in species richness (ranging from 11 to 13,720 species with a median of 1704), we used the Simpson similarity index for taxonomic similarity (Eq. 1)[49], which is largely insensitive to species richness:[50]

$$SimTax = 1 - \frac{\min(b, c)}{a + \min(b, c)} \quad (1)$$

Here $a$ is the number of species common to both regions, $b$ is the number of species that occur in the first region but not in the second and $c$ is the number of species that occur in the second region but not in the first[51]. Likewise, we calculated the Simpson phylogenetic similarity index as phylogenetic similarity (Eq. 2) as implemented in the R package 'betapart' (version 1.5.1)[52]:

$$SimPhyl = 1 - \frac{\min(B, C)}{A + \min(B, C)} \quad (2)$$

Here $A$ is the total length of the phylogenetic branches in the phylogenetic tree that are shared by the species of both regions, $B$ is the total length of the phylogenetic branches that are shared only by the first region and $C$ is the total length of the phylogenetic branches that are shared only by the second region[51]. To quantify changes in similarity due to naturalization of alien species, we calculated the degree of homogenization $H$ (or differentiation, see below) for each pair of regions as

$$H = ln \frac{Sim_{native+naturalized} + 0.001}{Sim_{native} + 0.001} \quad (3)$$

A small value of 0.001 was added to both similarities to avoid infinite values. A positive log-response ratio indicates homogenization (i.e., increased floristic similarity between two regions), and a negative one indicates differentiation (i.e., decreased floristic similarity between two regions). As an alternative to the Simpson similarity index, we also calculate the Sørensen similarity index, which additionally takes into consideration the nestedness of the floras in the paired regions[51]. As the results were not sensitive to the choice of similarity indices (Supplementary Fig. 14), we focused our analyses on the Simpson similarity index.

To quantify phylogenetic similarity, we used a phylogenetic tree including all angiosperms with accepted names in The Plant List (Supplementary Fig. 2). The tree was developed based on the mega phylogeny of Smith and Brown[53]. We added missing species ($n = 71,124$, of which 733 are naturalized in other regions) with

their accepted names in The Plant List to the root of their genus or families. For details on the development of the phylogenetic tree, see ref. [47].

### Quantification of geographic distances and climatic distances.
We calculated the pairwise geographic distance between regions as the distance between their geographic centroids using the R package 'geosphere' (version 1.5-10)[54]. We also calculated the nearest distance between the geographic borders of regions. However, since the distances between geographic centroids are highly correlated with distances between region borders ($n = 216,153$, $r = 0.996$, $P < 0.001$), we only used distance between region centroids in our analysis.

We quantified the pairwise climatic distance between regions as the distance between their positions in multidimensional climate space. We extracted all 19 biologically relevant variables of temperature, precipitation and seasonality from the WorldClim database[55] (Supplementary Table 1) at a resolution of 2.5 arc-min. As some of these bioclimatic variables are highly correlated, we first conducted a PCA on them, and used the first three principal axes, which are orthogonal to each other, as new climatic descriptors. Many of the bioclimatic variables had a skewed distribution and varied greatly in magnitude. Therefore, before including them in the PCA, we transformed each variable to be as approximate to a normal distribution as possible (Supplementary Table 3) using the R package 'normalizer' (version 0.1.0)[56]. We then scaled them to have a mean of zero and a standard deviation of one. For each 2.5 arc-min grid cell, we obtained its scores along the first three principal component axes, which in total explained 85.9% of the variance in the original bioclimatic data (Supplementary Fig. 15a). We then calculated the climatic centroid of cells extracted for each region within the three-dimensional climatic space, and quantified climatic distance between two regions as the Euclidean distance between their climatic centroids. In addition, as the first principal component axis (PC1) was mainly related to temperature variables, and PC2 mainly to precipitation variables (Supplementary Fig. 15b), we also calculated distances for PC1 and PC2 separately.

### Compiling data on administrative relationships.
To assess current and past administrative relationships among the 658 regions, we first assigned each region to its current country, and then compiled data on the past colonial relationships between the 110 countries represented by our regions. For colonial relationships, we only considered the period after 1492 (i.e., the year of discovery of the Americas by Columbus), which is the time period during which the major introductions and naturalizations of alien species happened[57]. As basis for our current and past administrative relationship dataset, we used the TRADHIST dataset[58]. Since TRADHIST is not comprehensive and describes only the administrative relationship between countries or states in the last two centuries, we added data on colonial relationships for the main colonial empires as listed in the Wikipedia article 'Colonial empire' (https://en.wikipedia.org/wiki/Colonial_empire). We next extracted data on relationships of dependent territories, which are defined as territories that do not possess full political independence as a sovereign state but remain politically outside the controlling state's integral area[59] (e.g., Guam [USA]) from the Wikipedia article 'Dependent territory' (https://en.wikipedia.org/wiki/Dependent_territory).

We were able to group administrative relationships of each pairwise combination of the 658 studied regions into three main categories: (i) same country: the two regions are currently part of the same country (e.g., California and Texas, both part of the USA); (ii) dependency relation: the two regions are either currently dependent territories or past colonies of the same country (e.g., Hong Kong and Ireland, both were historically colonies of the UK), or the two regions belong to two different countries of which one is or was the dependent territory or colony of the other (e.g., Guam [USA] and England; California [USA] and England); (iii) no administrative relation: the two regions have no current or past administrative relationships.

### Assessing the association of the change in floristic similarity with geographic distance, climatic distance and administrative relationship.
To describe the non-linear relationships of taxonomic and phylogenetic floristic similarities ($SimTax_{native}$, $SimTax_{native+naturalized}$, $SimPhyl_{native}$, $SimPhyl_{native+naturalized}$) along the gradients of geographic distance and climatic distance, we fitted single-predictor log-binomial generalized linear models (GLMs) following ref. [17]. The intercept of the model with geographic distance as the predictor was fixed at 1, assuming complete similarity at a distance of 0 km. Following ref. [16,17], we calculated and compared the halving distance (i.e., the distance at which a given similarity value is predicted to have decreased by 50%) of each of the four similarity indices.

To statistically test how changes in taxonomic and phylogenetic similarities (i.e., the degree of homogenization or differentiation) between two regions vary with geographic distance, climatic distance, administrative relationship, and their interactions, we used multiple regression on distance matrices (MRM)[60]. To account for non-independency of data points, caused by use of each region in multiple region pairs, the statistical significance of the MRM-model-coefficient estimates was assessed by comparing them to the null distribution of the coefficient estimates[60]. The latter was produced by simultaneously shuffling the rows and columns of the response matrix (i.e., degree of homogenization between regions). This permutation was done 999 times (see Ref. [60] for an example of the permutation process). As the MRM models assume linear relationships, we additionally used Generalized Additive Models (GAMs) to check whether any possible nonlinear effect of geographic distance or climatic distance would change our conclusions from the linear MRM model. Since the results of both models were qualitatively consistent with regard to the main effects, we present only the results of the linear MRM model in the main text, and the GAM results and some deviations with regard to the interaction effects in Supplementary Fig. 5. To assess whether the effects of climatic distance were mainly due to temperature or precipitation variables, we ran additional MRM analyses in which we replaced the single climatic distance measure with distances based on PC1 and PC2, respectively (Supplementary Fig. 15b).

### Assessing homogenization hotspots and associations with characteristics of the regions and their floras.
To assess where the global homogenization hotspots are, and how homogenization values depend on characteristics of the regions and their floras, we first averaged for each region all of its pairwise homogenization values $H$. We then extracted a set of variables characterizing the regions and their floras, including the species richness and phylogenetic diversity of native and naturalized floras, the proportion of endemic species, the donor score of the region (calculated as the average number of non-native regions each native species is naturalized in), the size of the region, and whether the region is an island or a mainland region. We then did a linear multiple regression of the average degree of homogenization of the regions on the variables characterizing the regions and floras. Of these predictors, species richness and phylogenetic diversity, the donor score, and the region size were log-transformed to increase their linear association with homogenization. To compare their relative importance, predictors were scaled to have a mean of zero and a standard deviation of one. We reduced spatial autocorrelation in the residuals of the linear model by adding a spatial auto-covariate that incorporates a matrix of longitude and latitude coordinates of the centroids of the regions with the R package 'spdep' (version 1.1-3)[61] (Supplementary Fig. 16). All data extraction, statistical analyses and figures were done using R version 4.1.0[62].

**Reporting summary.** Further information on research design is available in the Nature Research Reporting Summary linked to this article.

## Data availability
The core datasets of this study are available at https://figshare.com/articles/dataset/Data_and_Code_for_Yang_et_al_The_global_loss_of_floristic_uniqueness/14991624. These include (1) similarity and degree of homogenization between pairwise regions, (2) geographic distance, climatic distance, and the administrative relationship between pairwise regions, (3) average degree of homogenization of each region, (4) characteristics of regions and their floras, including the richness and phylogenetic diversity of native and naturalized species, the proportion of endemic species, the donor score, the size of the region, and whether the region is an island or a mainland region, and (5) phylogeny of the seed plants with accepted names in TPL and a dataset describing the taxonomic group (e.g., family and order) of each species and how the species was added to the phylogeny if it was initially missing from the phylogeny. The GloNAF database together with the shapefile that was used to produce the maps have been published in a data paper[45], and the most recent version is available upon request. An R package making data from the GIFT database publicly available is under development. The original phylogeny ALLMB is available at https://github.com/FePhyFoFum/big_seed_plant_trees/releases. The dataset TRADHIST can be downloaded at http://www.cepii.fr/CEPII/en/bdd_modele/presentation.asp?id=32. The Plant List database can be accessed at http://www.theplantlist.org/1.1/browse/A/. The database Worldclim can be accessed at http://www.worldclim.com/version2. A detailed description of the databases and their access approaches can be found at Supplementary Table 4.

## Code availability
One document including the R codes of the main analysis, and the relevant output of the codes is also available at https://figshare.com/articles/dataset/Data_and_Code_for_Yang_et_al_The_global_loss_of_floristic_uniqueness/14991624.

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

## Acknowledgements

The authors would like to thank the KUZ Herbarium of the Kuzbass Botanical garden, the herbarium TAA from Estonian University of Life Sciences, and all other data contributors of GloNAF and GIFT. QY, AS, MW, and MvK acknowledge support from the German Research Foundation DFG (grants KL 1866/9-1, FZT 118 202548816 and 264740629). PP and JP were supported by EXPRO grant no. 19-28807X (Czech Science Foundation) and long-term research development project RVO 67985939 (Czech Academy of Sciences). HS acknowledges support from Belmont Forum-BiodivERsA project AlienScenarios through the national funders German Federal Ministry of Education and Research (BMBF; grant 16LC1807A). FE and BL acknowledge funding from the Austrian Science Foundation FWF (grants I2086-B16 and I 4011-B32). NF acknowledges the support by the grant Fondecyt No 1181688. ELHG acknowledges the support from the São Paulo Research Foundation (FAPESP grant #2012/06005-1).

## Author contributions

Q.Y., A.S., and M.v.K. designed the study. Q.Y., performed data extraction with key contributions from A.S., Q.Y., and M.v.K., analyzed the data and led the writing, with key contributions from T.F. P.W., T.F., Z.Z., H.S., and A.S. contributed important ideas. W.D., F.E., H.K., C.K., B.L., R.P., J.P., P.P., M.W., A.E., N.F., E.G., J.K., P.K., T.K., M.N., K.N., J.V., J.W., A.Z., and E.Z., contributed data. All authors contributed to discussion and writing.

## Funding

## Competing interests

The authors declare no competing interests.
