## [Peer Review File · Nature Communications]

Reviewers' Comments:

Reviewer #1:

Remarks to the Author:

This manuscript is grounded in a dataset that is rich in analytical possibilities. The authors have undertaken a large number of analyses some of which are well constructed and warranted, and others which are underdeveloped in their current form. Below I provide some suggestions which I hope will assist the authors in seeing how their work is interpreted by the reader. There are many opportunities for refinement. I have numbered my suggestions so that they can be easily distinguished and actioned.

SUBSTANTIVE COMMENTS

1. The manuscript needs a figure which summarises the conceptual framework being presented in lines 95-107. This text was difficult to parse, but essentially amounts to the different ways in which naturalised taxa may overlap between regions and the consequences this may have through both homogenisation and 'heterogenisation' on floristic diversity. With this in place, the various ideas being tested will be clearer to the reader. I am not convinced that heterogenization is going to be a useful term to introduce to the literature. I understand that there is a desire to present the opposing term, but perhaps simpler language which lends from the established vernacular around diversity (e.g. similarity, commonness etc.) may tie this work into some existing, broader concepts.

2. Overall, the use of a series of metrics to assess anthropogenic pressure on regions seemed out of place in the manuscript and distracted from the key messages about floristic exchange. I would prefer to see these analyses removed from the manuscript – particularly the two outlined below – to leave more space for nuanced discussion of the regional difference in floristics driven by biogeography and past trade/colonial relations. The manuscript attempts too much by having these elements included and the connections to the main topic is tenuous. For instance:

a. The use of the UNDP Human Development Index (HDI) as a measure of anthropogenic pressure is problematic. It is unclear how the three tenets of the HDI as stated: "living a long and healthy life, being knowledgeable and having a decent standard of living" approximate an anthropogenic pressure. High standard of living is not necessarily compatible with anthropogenic pressure – in fact, one may equally argue that societies with a high standard of living have more opportunity to invest in sustainable approaches to resources extraction and other practice which threaten plant diversity. This index is a poor approximation of the pressure facing floristic assemblages from naturalisation – far too peripheral to be included as an explanatory factor in the analyses presented.

b. The use of a 16 year-old metric of biodiversity intactness will give a poor approximation of the contemporary state of biodiversity loss globally. Decline in biodiversity is rapid and dynamic, with areas emerging as hotspots of both destruction and protection since 2005. This inclusion distracted from the more grounded narrative around the role of biogeography and trade – two factors widely recognised as barriers and drivers to biotic exchange. Removing the more peripheral elements of the analysis will make for a stronger manuscript. This will also allow for more space to explore the consequences of homogenisation from a conservation perspective, a theme which is currently underdeveloped (see below).

3. The manuscript focusses heavily on how floristic composition has been affected by the introduction process, however I wanted to see more evidence that the naturalisation of species has (i) led to the formation of self-sustaining populations in each region in contrast to simple introduction; and (ii) has any significant consequences for the function of vegetation across regions. The key missing element is abundance. While some regions may have greater richness of naturalised plants, this does not translate to an effect on the ecology of the majority of systems. The presence of a large cohort of naturalised species does not translate to a homogenised flora unless there are data on abundance to demonstrate that these introduced taxa have been integrated into the floristic assemblages of regions in sufficient number to alter the ecology of an ecosystem. The authors did not acknowledge the important role of species abundance in shaping

the process of homogenisation in the flora of the regions they have investigated. While it may be true that the catalogue of plant names in a region may change through naturalisation the consequences of this mixing are entirely governed by the abundance of the species in the landscape. I saw no attempt to address this obvious and important caveat, rather (Line 287) states that hotspot regions have 'lost most of their floral uniqueness' and this is profoundly overstated. There was no discussion of how a large naturalised species pool in a region can contribute to higher rates of plant invasion (i.e. species which reach high abundance).

4. Using a spatial proxy to test a climatic hypothesis. The hypothesis about climatic suitability is not tested directly, instead the authors group regions into 'tropical' and 'non-tropical' and then test for differences. This seems like a weak test of the existing knowledge about the role of specific factors such as rainfall and temperature in the naturalisation process. Global gridded climatic data should be used to estimate the climatic niche of each region, or even better, each species (both mean/median and range) and this data used in assessing the relationship between climate and homogenisation. This matters for several reasons:

a. because tropical and non-tropical regions differ in several ways which are not related to climate. Most importantly, there is a strong gradient in species richness with latitude leading to higher numbers of plant species in the tropics than the temperate zone. These differences may shape how homogenisation proceeds in the zones and has not been independently separated from the effect of climate if not using more direct measures.

b. Within both tropical and non-tropical zones there is a large amount of regional variation in factors such as topography which shape the composition of floristic assemblages. For instance, tropical cloud forest and lowland rainforest may differ markedly within a region in their floristic composition.

Without significant revision of the treatment of climate the manuscript misses an opportunity for making significant insights into what might shape the process of homogenisation. Further, the presentation of the climatic results is provided across disparate paragraphs with some discussion of administrative relationships in between. This structure was difficult to follow and repetitive and detracted from the overall message of the manuscript.

5. Reporting of results: I was surprised to see main findings lacking reporting of statistical tests e.g. line 219-22. In this instance, the results are described qualitatively using language like 'slightly stronger increase'. How is the reader to assess the strength of the relationship without the presentation of the statistical data comparing slopes? The only reference made here is to Figure 3 where multiple slopes are provided and readers are asked to infer that one relationship is slightly stronger than another. It seems some explicit test comparing slopes is warranted.

6. I would recommend revisiting the data from island regions relative to regions with land borders as a potentially rich source of information on how biotic homogenisation proceeds. Geographic isolation provides islands with a natural barrier to biotic transfer. Patterns should be reinforced in these regions, relative to regions sharing land borders and I imagine this would be a testable approach for the dataset. The Australasian (line 254) and oceanic island (line 258) results which are [presented reinforce this idea, without explicitly testing if biogeographic isolation leads to stronger homogenisation – this seems like another missed opportunity.

7. Summary: The summary overstated the need for the manuscript, attempting to present a wider knowledge gap than appears in the literature on the causes and consequences of biotic homogenisation. For instance, the statement around how 'drastically' the process of naturalisation has led to changes in floristic diversity is 'still missing' from the literature is excessive. The analyses presented do provide a wider perspective on the topic, but are by no means the first to be undertaken (and this is acknowledged by the authors in the results of the manuscript which point to islands being well-known hotspots of biotic homogenisation – one of their key findings). Work has been completed on this topic since the 1990's and several of the papers have attracted much attention.

MINOR COMMENTS

Line 130: it is not immediately clear to the reader why climatic similarity may affect the naturalisation process. This needs stronger justification.

Lines 130-134: although an expectation is given, the statement about administrative connections needs more nuance to make it a clear hypothesis. Perhaps adding the word 'increased' would help -instead of discussing the 'degree' of homogenisation?

Line 137: If retained, the human development index should be correctly referred to as the 'UNDP Human Development Index (HDI)' in the Introduction.

Line 176: "prerequisite" is too stronger word here – soften the language. Climate is one barrier to introduction for some species, but many species will thrive outside their climatically suitable habitat through niche shifts, or through husbandry or via genetic founder effects which resulting from the introduction of particular populations. Most simply stated, the hypothesis here is that climatic similarity predicts biotic homogenisation between region pairs.

Line 199-200: it is not clear to the reader what the connection is between the analysis of multiple region pairs allow for testing of 'administrative relationships'. The definition of administrative relationships is not adequately developed making the connection between these two data sources seem like a non-sequitur.

Lines 203-209. This is a much stronger way to summarises the findings, by showing the effect of naturalisation on the rate of spatial decline in taxonomic/phylogenetic similarity. This information would be better placed earlier in the discussion of the results as it is a more intuitive way of conveying the results which are about geographic distance. However, one thing to note is that the paper deals with naturalisations, not invasions and the use of this term is misleading.

The writing style and expression requires attention throughout. For instance, sentences are often long and as a result hard to comprehend. Examples include lines 112-116, or lines 134-140 which is six(!) lines long and full of different concepts. The expression is often quite 'loose' in that bold statements are made without sufficient nuance (e.g. line176 about climatic suitability) or sentences are poorly constructed (e.g. line 178 ends with 'will be'). I understand that Nature format requires brevity, but this should not come at the expense of clarity.

Line 245: eucalyptus species, not trees – the results are not at an individual level, but at a species level.

Line 270: I wanted a little more information from this finding about small island having high phylogenetic similarity to native floras of other regions. Seems to me that this result will be due to the processes involved in island formation, which may cut off areas previously connected to mainland regions (e.g. New Caledonia and Australia).

It would be nice to see fewer analyses and more depth in the interpretation of what is offered. For instance, here's a significant amount of inference being made here about how colonial histories may shape trading of species. While I do not disagree that this is a potential pathway for floristic exchange, I would have liked to have seen a stronger suite of predictors to really tease out this result.

Reviewer #2:

Remarks to the Author:

The study by Yang et al. provides a fascinating and important global-scale analysis of the impact of plant naturalizations to floristic homogenization – that is, the loss of uniqueness in composition among evolutionarily and historically distinct biogeographic floras. At first, I was skeptical of the claim in the Abstract that information on the contributions of non-native plants to biotic homogenization is "still missing," as there are many studies on biotic homogenization in the context of invasions and the concept is far from understudied. However, as I read on, the analyses presented are indeed very unique, which have similarly been done at smaller scales but only

speculated at the global scale. The authors find: 1) Homogenization is occurring at global scale, with naturalizations overwhelmingly increasing taxonomic and phylogenetic similarity between distant floras – the expected decay in similarity with distance is decreasing due to anthropogenic mixing of floras; 2) Homogenization is more severe between broadly climatically similar regions; 3) Homogenization is greater between regions that were historically or currently connected through administrative relationships (eg. colonialism). There are many interesting sub findings within those conclusions as well, including interesting island-mainland differences. Though the results are not entirely surprising (though some aspects are unexpected), this study provides a quantification of the phenomenon of anthropogenic mixing of regional floras. It has the potential to be highly cited and to spur additional research on the multidisciplinary study of causes behind the patterns, and their past, present, and future impacts. Isolated floras are no longer isolated, and the analyses in this paper leverage impressive datasets on both native and naturalized floras to quantify the impacts of floristic interchange in the Anthropocene.

I am unclear on the impact of the region size on the results. I think this concern is minor, as I suspect this is my confusion. The authors combine many checklists of native and naturalized species at various scales to create contrasts among 658 regions. However, as stated in Methods (L315), the regions represented in the GloNAF database are hugely variable in area, ranging from 0.03 sq. km to nearly 7 million sq. km. However, I am unsure on the criteria for merging these regions in the 658 regions used in the analysis. Were regions defined based on biological or practical criteria? What is the range of area in these regions? Are the metrics used in the contrasts sensitive to size of regions? I would expect that both the geographic and climatic distances used in analyses to be particularly sensitive to region size disparities between the contrasts, given that the distances are calculated based on geographic and climatic centroids (L373-390). Further, how were regions "...judged to include more than 50% of the naturalized taxa for that region" (L329)? The authors state Simpson similarity metrics used are robust to species richness, but I am unclear if results in this context are also insensitive to region size. Maybe it does not matter. Ideally contrasts of regional floras could be defined based on some a priori biogeographic (such as Takhtajan's floristic regions or similar) rather than more arbitrarily defined regions.

I applaud the authors for beautiful and easy to understand figures. There is a lot to take in!

Global invasion patterns are notably asymmetric with some regions more likely to be donors than recipients, as has been well shown regionally in several studies and globally by a subset of the current authors (van Kleunen et al. 2015 Nature). This notion of asymmetric species exchanges could be important to include in the introduction or conclusion in the context of homogenization. Further, I am curious: are "predominantly donor" regions less likely to be homogenization hotspots?

I assume the GLONAF dataset does not comprehensively capture "invasive" species from "naturalized" species. It may be interesting to speculate on the role invasive species have on realized homogenizations, as many naturalized species are limited in their extent and/or abundance.

A few minor comments:

L146: An equation here (or in Methods) to show the components and calculation of the similarity metrics would be helpful. A statement that greater positive values indicate homogenization, while more negative values indicate heterogenization.

L188: Please give test statistic with p value.

L226: "...contain signals of colonial history dating back at least 500 years." I think some elaboration is needed, as I was unclear where this 500 years originated. The oldest administrative connection in dataset?

L264: The finding of heterogenization on some islands was unexpected and interesting. The reasoning of higher phylogenetic similarities with mainland floras makes sense.

Figure 2: Please add sample sizes for boxplots in the caption and/or figure and/or supplement.

Figure 3: Please define meaning of "dependency" in caption.

Figure 4: Perhaps add that negative values indicate heterogenization in the caption for clarity.

The code seems very well annotated and accessible, which was no easy task given the scope of the analyses and datasets. Only a gentle suggestion - it could also perhaps be permanently archived to allow static dataset and versioning.

A lingering thought, perhaps best for future work:

It would be fascinating to also analyze functional homogenization – how is trait similarities/diversities changing? Does phylogenetic homogenization match functional homogenization? GIFT includes traits, so perhaps this was already part of the research team's agenda.

Mason Heberling
Carnegie Museum of Natural History

Reviewer #3:

Remarks to the Author:

SUMMARY

This paper by Qiang Yang and colleagues aims at quantifying both: the native floristic similarity between regions of the world, and whether this similarity is affected by the presence of naturalized alien species. They carry their analysis at a global spatial scale, on 60% of all native angiosperm species and >10'000 naturalized alien angiosperm species. The effects of alien species on (taxo and phylo) similarities between regions is then analysed in regards to several factors including geographic distances, climatic differences, administrative histories, levels of human developments, and biodiversity intactness. They found that the presence of naturalized alien plants generally increases both taxonomic and phylogenetic similarities between regions.

WRITING & PRESENTATION

The manuscript has been carefully written and I enjoyed reading it. It is also well organized and the figures are nice.

CONTEXT & ORIGINALITY

I agree with the authors that an analysis of (taxonomic and phylogenetic) similarities at such large spatial and taxonomic scales is novel for plants species and potentially interesting. However, under its current form, the manuscript could not convince me about how the results are important for our understanding of the threat posed by invasive species, and whether their methodological choices are the most relevant ones to understand invasion-driven biotic homogenization (see my major comments below).

GENERAL COMMENTS

I have 3 general concerns that I think should be answered, they include: (1) the need to clarify the general aims and hypotheses of the study, (2) the need to clarify the link between invasion-driven homogenization to the other tested factors, as well as the methodological choices associated, and (3) a series of methodological questions. I detail them below.

(1) The need to clarify the general aims and hypotheses of the study.

I think it is important to explain the reader: (i) why invasion-driven homogenization is a threat to biodiversity and not a good thing (as this kind of homogenization is about species gains, and not species loses)? And (ii) why one should care about phylogenetic uniqueness of a flora?

(1.i) L. 80 (3rd sentence in the intro): "Importantly, [alien species] also change patterns of biotic uniqueness or ^{SEP}distinctiveness of those regions compared to others, with potential ecological and ^{SEP}evolutionary consequences (Ref 4, 16)". This is a key sentence as it sets the rational and motivation for the whole study: why it is important to understand invasion driven homogenization.

However, the authors never explain why it is "important" and what are those "potential ecological and evolutionary consequences". This lack of explanation questions the actual relevance of such analyses. In fact, one can genuinely wonder why adding more species to a region has important eco and evo consequences, and whether these consequences would be harmful to the ecosystem. I guess that one part of the answer is that naturalized species may harm in some way native species, and thus can lead to local biodiversity reduction (argument proposed in their ref 16), but this is not really what is investigated here (as there is no data on native species range reduction for example).

(1.ii) Why should people care about phylogenetic uniqueness of a flora? The phylogenetic aspect of uniqueness is mentioned L.109, but without an explanation about why it is interesting to understand this aspect. What is the argument? What means "unique evolutionary history" (I understand the words, but not the ecological implications), and why should we account for it in conservation (ie. is it a proxy for functional uniqueness? Or is there something else?)

Additionally, some "expectations" for the different analyses of this work are listed in the introduction (e.g. L.134-140), but the rationale behind these expectations are not explained to the reader. Please add these explanations, and indicate whether these are novel speculations or results already found in the literature.

Don't get me wrong, I am sure the authors have plenty of good scientific motivation for their work, but it is essential that they make these motivations explicit to the reader so that one can take the measure of the value of this work and its implications for our understanding of invasion and protection of biodiversity.

(2) The need to clarify the link between invasion-driven homogenization to the other factors, as well as the methodological choices associated:

I did not really understand the usefulness of linking invasion-driven homogenization to the other factors, please clarify your motivations (or consider removing these analyses). Here are a few examples of my lack of understanding:

(2.i) Why is it interesting and important to link homogenization to geographic distances? What I understood is that we learn that invasive species are less sensitive to environmental barriers than native species. I do not really understand the novelty of this finding (it is somewhat in the definition of invasion). If there are other conclusions that are drawn from this analysis, please make them more explicit. Otherwise, I suggest removing it.

(2.ii) L. 63-65 (abstract): "floristic homogenization was greater between regions with current or past administrative relationships than for regions without, indicating that trade, transport and colonial history facilitate floristic exchange". I did not see what was new in this finding: both trade and transport are the means by which invasions occur, by definition. The one new thing for me here is the colonial history part, but then it is not really a driver of invasion per se, rather an indicator of the intensity of transport and trade, right? Please clarify this point. Additionally, if the idea is to show that colonial history has favored invasion, most likely via increased transport and trade, wouldn't it be better to use a more direct test of this than going through homogenization measures? (e.g. testing whether species probability to invade a region depends on whether it is native to another region of the same colonial empire; or use SEMs with all info together: colonial history, trade, transport, invasion; or any relevant alternative).

Similar concerns can be applied to the other factors, please take the time to clarify the text in this regard. [e.g. L.276-278: "Taxonomic homogenization decreased with biodiversity intactness, indicating that regions in which there is less loss of native biodiversity have also suffered less from plant invasions." I do not understand why you need to go through analyses of beta-div to test such hypothesis. Same reasoning works for the following (L. 281-282) about the link between invasions and human populations.]

(3) Methodological questions and concerns:

(3.i) Do you need null models to quantify significant alien species effects?

It seems to me that there could be a mechanical effect of species richness on the change of similarity when considering only native species or native+invasive species. Isn't it correct that strong homogenization by alien species is much easier for communities with few native species than for species rich communities, as invaders can quickly become a large proportion of the dissimilarity between regions? If so, could this effect be controlled by the use of a null model, testing "are the observed effects of alien species on region similarities, greater or smaller than those expected in two regions of the same taxo richness but containing a random selection of

species?" (or any other more relevant null model if you think of a better one). The random expectations could be calculated from random assemblages constructed using the 'independent swap' algorithm that randomizes species co-occurrence but keeps constant the sample richness per community and frequency of occurrence of each species across all communities (Gotelli & Entsminger, 2003). Doing so would also help quantifying whether the alien species effect are "significant" or not.

Similarly, regarding the alien species effects on phylogenetic uniqueness, a null model could be used to quantify whether the invasive species are creating significantly more phylogenetic homogenization than expected by random addition of species (e.g. by randomization of the tips of the phylogeny?).

(3.ii) Also, I think that it would be important to understand a little better what drives the differences in similarities across region pairs. Are they driven by their level of endemism of native species and phylogenetic distinctiveness? Are there some effects of the region area on the results (cf. Barton et al 2013 GEB)? And is it linked to local native richness, alien richness, or alien:native ratio? Are most of the homogenization patterns driven by the most widespread invaders, while the heterogenization patterns are rather driven by the "narrow" range invaders (and if so is there a sampling bias toward wide-spread invaders in the dataset, with which consequences for the results?)?

(3.iii) Finally, I think that it is necessary to discuss the results in regard to the potential weaknesses of the dataset. Typically, are the missing species (40% of angiosperms missing) randomly spread in the phylogeny, and where are the alien species in these "data holes"? This would help assessing whether the phylo-similarity and homogenization can be biased toward over- or under-estimations. Also, some homogenisation cold spots are in Asia, central America & Amazonia, central Africa (Fig. 4); are these influenced by the data quality there?

SPECIFIC COMMENTS

References: Within the first 15 references I counted 10 references including at least one co-author of this manuscript. I understand that many of the authors have been working on invasions for a long time now and have published a lot of good work on related research areas. However, I believe it would bring a little more credit to the generality of this work if they would cite a little less of their own papers.

L.58: "significant" this is misleading as it somehow suggests $p\text{-value} < 0.05$ while there was no test here.

L.100. "the size 'OF' the different sets.." typo?

L.167: I do not understand the argument here, if species are alien to both regions then they can only homogenize them (as explained in the introduction), no? why do you here expect heterogenization?

L. 172-174: "So, the decrease in floristic similarity of two regions is not solely driven by the exchange of species that are native to one of the two regions, but also by alien species from elsewhere that have invaded both regions." This is not really a finding of your study, but rather a description of how you measure homogenization. Or did I misunderstand the argument here?

L. 213-217: "... the degree of taxonomic homogenization increased more rapidly with geographic distance for region pairs belonging to the same country than for other region pairs (Fig. 3c, Supplementary Figs. 3&5d). This most likely reflects facilitated spread of an alien species within a country after it has become established in one of the subregions." As mentioned in the general comments, if that is the question, then why don't you just analyze the probability of naturalization as a function of administrative borders, or colonization history?

L.226-227: "floristic homogenization contains signals of colonial history dating back at least 500 years". I don't see how you can say that the signal dates from 500 years here. Maybe the signal you found is just driven by what happened the last 100 years? Either remove this statement or test explicitly the dating (e.g. use different colonial subsets of different time periods).

L.240-243: sentence unclear, please breakdown or clarify. (this one paragraph reads less well than the others, with a lot of terms like "increase" "decrease" "similar" "dissimilar" "distance" "distant" "nearby", you may want to simplify it).

L.270-273: I did not understand this argument, please clarify.

L. 349: I calculate 215,824 pairwise combinations of different regions ($(658 \times 658) - 658 = 215,824$), and not 216,153 as indicated.

L368: please add the number of alien species that are part of these missing species in the

phylogeny (to clarify that the aliens were not particularly badly resolved).

L.435: please add a ref for this null distribution of coeff approach?

- Please add the formula of the Simpson similarity index used (in the sup mat if your are limited by space in the main text), and a sentence to explain why you chose this single index to describe similarities across regions (e.g. and not a pair of turnover & nestedness components)? As it is written in the method section now it seems that this index choice is due its insensitivity to species richness differences across regions, but a null model approach could be used to solve this issue, no? is there another reason?

- Maybe worth adding a sentence in the methods or discussion explaining what are the implications of using artificial (here political units such as states, provinces, counties) rather than natural biogeographical regions when quantifying biotic homogenization? (even though I understand that the data per biogeo region is not available)

Responses to reviewers (NCOMMS-21-03266)

We would like to thank you and all reviewers for the constructive and thoughtful comments, which have helped to improve the quality and potential impact of the manuscript. In summary, the main changes are:

- We have restructured the manuscript. Now, we have separate Results and Discussion sections. We also restructured the Introduction to clarify the general aims and hypotheses better.
- We have removed the analysis of how the mean extent of a region's homogenization is related to indices of anthropogenic influence, and replaced it with an analysis of how the mean extent of a region's homogenization is related to area of the region, native species richness, naturalized species richness, degree of endemism and whether the region is an island or mainland region.
- We deepened the analysis of the importance of climatic similarity by calculating the similarity in PC1 (reflecting similarity in temperature) and similarity in PC2 (reflecting similarity in precipitation).

We respond in detail (*in blue text*) to the comments of each reviewer below.

Sincerely,
Qiang Yang (on behalf of all authors)

Reviewer #1 (Remarks to the Author):

This manuscript is grounded in a dataset that is rich in analytical possibilities. The authors have undertaken a large number of analyses some of which are well constructed and warranted, and others which are underdeveloped in their current form. Below I provide some suggestions which I hope will assist the authors in seeing how their work is interpreted by the reader. There are many opportunities for refinement. I have numbered my suggestions so that they can be easily distinguished and actioned.

SUBSTANTIVE COMMENTS

1. The manuscript needs a figure which summarises the conceptual framework being presented in lines 95-107. This text was difficult to parse, but essentially amounts to the different ways in which naturalised taxa may overlap between regions and the consequences this may have through both homogenisation and 'heterogenisation' on floristic diversity. With this in place, the various ideas being tested will be clearer to the reader. I am not convinced that heterogenization is going to be a useful term to introduce to the literature. I understand that there is a desire to present the opposing term, but perhaps simpler language which lends from the established vernacular around diversity (e.g. similarity, commonness etc.) may tie this work into some existing, broader concepts.

RESPONSE: We thank the reviewer for the helpful suggestion of including a figure of the conceptual framework to navigate the reader through the many different ways in which naturalized taxa may overlap between regions and the consequences these

different ways could have on floristic uniqueness. Therefore, we altered the previous Supplementary Fig. 1 to more clearly illustrate the different ways that naturalization can change the similarity of two floras and now moved it to the main manuscript (Fig. 1 in the revised manuscript). In addition, we added to the supplement an extended version of this figure in which we illustrate the changes in the taxonomic and phylogenetic similarity between two regions for different naturalization scenarios (Supplementary Fig. 1).

We replaced the term “heterogenization” with “reduced similarity” or “differentiation” (as also used in Winter et al. 2009, PNAS 106:21721-21725) throughout the text. Moreover, we now generally use the neutral term ‘change in (floristic/taxonomic/phylogenetic) similarity’ instead of ‘degree of homogenization’ to make it clear that a naturalization event does not necessarily result in homogenization.

2. Overall, the use of a series of metrics to assess anthropogenic pressure on regions seemed out of place in the manuscript and distracted from the key messages about floristic exchange. I would prefer to see these analyses removed from the manuscript – particularly the two outlined below – to leave more space for nuanced discussion of the regional difference in floristics driven by biogeography and past trade/colonial relations. The manuscript attempts too much by having these elements included and the connections to the main topic is tenuous. For instance:

a. The use of the UNDP Human Development Index (HDI) as a measure of anthropogenic pressure is problematic. It is unclear how the three tenets of the HDI as stated: “living a long and healthy life, being knowledgeable and having a decent standard of living” approximate an anthropogenic pressure. High standard of living is not necessarily compatible with anthropogenic pressure – in fact, one may equally argue that societies with a high standard of living have more opportunity to invest in sustainable approaches to resources extraction and other practice which threaten plant diversity. This index is a poor approximation of the pressure facing floristic assemblages from naturalisation – far too peripheral to be included as an explanatory factor in the analyses presented.

b. The use of a 16 year-old metric of biodiversity intactness will give a poor approximation of the contemporary state of biodiversity loss globally. Decline in biodiversity is rapid and dynamic, with areas emerging as hotspots of both destruction and protection since 2005. This inclusion distracted from the more grounded narrative around the role of biogeography and trade – two factors widely recognised as barriers and drivers to biotic exchange. Removing the more peripheral elements of the analysis will make for a stronger manuscript. This will also allow for more space to explore the consequences of homogenisation from a conservation perspective, a theme which is currently underdeveloped (see below).

RESPONSE: We followed the suggestion of the reviewer to remove the assessment of anthropogenic impacts on changes in regional floristic uniqueness. We, however, kept the maps of the mean change in similarity for each region and now use these mean values of the regions to analyse the effects of region area, native species richness, naturalized species richness, donor role, degree of endemism and insularity on the changes in taxonomic and phylogenetic similarity. This was done in order to address some of the other comments of the reviewers (see below).

3. The manuscript focusses heavily on how floristic composition has been affected by the introduction process, however I wanted to see more evidence that the naturalisation of species has (i) led to the formation of self-sustaining populations in each region in contrast to simple introduction; and (ii) has any significant consequences for the function of vegetation across regions. The key missing element is abundance. While some regions may have greater richness of naturalised plants, this does not translate to an effect on the ecology of the majority of systems. The presence of a large cohort of naturalised species does not translate to a homogenised flora unless there are data on abundance to demonstrate that these introduced taxa have been integrated into the floristic assemblages of regions in sufficient number to alter the ecology of an ecosystem. The authors did not acknowledge the important role of species abundance in shaping the process of homogenisation in the flora of the regions they have investigated. While it may be true that the catalogue of plant names in a region may change through naturalisation the consequences of this mixing are entirely governed by the abundance of the species in the landscape. I saw no attempt to address this obvious and important caveat, rather (Line 287) states that hotspot regions have ‘lost most of their floral uniqueness’ and this is profoundly overstated. There was no discussion of how a large naturalised species pool in a region can contribute to higher rates of plant invasion (i.e. species which reach high abundance). *RESPONSE: (i) We only used naturalized alien species, which by definition means that the alien species in our study have formed self-sustaining populations in the new region. We had mentioned this in the Abstract (L54-55) and Introduction (L76-77), but we now also repeat it in the Methods section (L444).*

(ii) We fully agree that it would be nice to have abundance data of all naturalized and of all native species in each region. Unfortunately, for most species and regions, such data are not available. However, because a global-scale quantification of homogenization is currently missing, we are confident that our work (though limited to analyses of regional floras) provides an important contribution to understanding how species introductions are changing global diversity patterns. Nevertheless, we agree that in order to judge the consequences of homogenization, data on the abundance of the species would be valuable, and we now mention this in the Discussion (L407-415). There, we also point out that McKinney & Lockwood (2005; In: Species Invasions – Insights into Ecology, and Biogeography, eds. Sax et al., pp 365-380) found that taxonomic similarity values based on presence-absence data only and taxonomic similarity based on abundance data were strongly correlated. They concluded: “species abundance and evenness apparently play a surprisingly small role in homogenization”.

4. Using a spatial proxy to test a climatic hypothesis. The hypothesis about climatic suitability is not tested directly, instead the authors group regions into ‘tropical’ and ‘non-tropical’ and then test for differences. This seems like a weak test of the existing knowledge about the role of specific factors such as rainfall and temperature in the naturalisation process. Global gridded climatic data should be used to estimate the climatic niche of each region, or even better, each species (both mean/median and range) and this data used in assessing the relationship between climate and homogenisation. This matters for several reasons:

a. because tropical and non-tropical regions differ in several ways, which are not

related to climate. Most importantly, there is a strong gradient in species richness with latitude leading to higher numbers of plant species in the tropics than the temperate zone. These differences may shape how homogenisation proceeds in the zones and has not been independently separated from the effect of climate if not using more direct measures.

b. Within both tropical and non-tropical zones there is a large amount of regional variation in factors such as topography which shape the composition of floristic assemblages. For instance, tropical cloud forest and lowland rainforest may differ markedly within a region in their floristic composition.

Without significant revision of the treatment of climate the manuscript misses an opportunity for making significant insights into what might shape the process of homogenisation. Further, the presentation of the climatic results is provided across disparate paragraphs with some discussion of administrative relationships in between. This structure was difficult to follow and repetitive and detracted from the overall message of the manuscript.

RESPONSE: In another analysis in our manuscript, we already did exactly what the reviewer is suggesting, i.e., we included climatic similarity, based on gridded climate data, as a predictor of the extent of homogenization (Fig. 3). To avoid that readers might miss this, we now present the results of this analysis before the results of the tropical vs non-tropical regions. We now also deepened this analysis by also running an additional analysis in which we replaced the single overall climatic similarity measure (based on the 19 WorldClim bioclimatic variables) with similarities based on the most important individual axes of a climatic PCA (PC1 and PC2), which largely represent similarities in temperature and precipitation, respectively. We present these new results in the supplementary information (Supplementary Figs. 6-7) and refer to them in the main manuscript (L193-199).

We fully agree that tropical and non-tropical regions do not only differ in climate but also in many other factors, and that these factors vary within each zone. Nevertheless, we think that most people will agree that two tropical regions or two non-tropical regions are more similar to each other than a tropical and a non-tropical region (be it with regard to climate, species richness, or some other factors) and that this is likely to determine the change in the floristic similarity between the two regions. We therefore strongly argue that the previous Fig. 2 and associated analyses provide a valuable complement to the more detailed climate analyses by intuitively showing that e.g., homogenization is most pronounced for distant regions in similar climatic zones. Therefore, we would like to keep the previous Fig. 2 (now Fig. 4 in the revised manuscript). However, we now emphasize more strongly that tropical and non-tropical regions do not only differ in climate (L291-293).

5. Reporting of results: I was surprised to see main findings lacking reporting of statistical tests e.g. line 219-22. In this instance, the results are described qualitatively using language like ‘slightly stronger increase’. How is the reader to assess the strength of the relationship without the presentation of the statistical data comparing slopes? The only reference made here is to Figure 3 where multiple slopes are provided and readers are asked to infer that one relationship is slightly stronger than another. It seems some explicit test comparing slopes is warranted.

RESPONSE: We thank the reviewer for pointing out this omission. We now added the P-value of the comparisons. Please, see L161, L164, L206, L211, L220, and L237. The main data analysis of this study involved three groups of regression models, including 1) GLMs regressing taxonomic and phylogenetic similarities to geographic distance and climatic distance between region pairs, 2) MRMs regressing taxonomic and phylogenetic homogenization to geographic distance and climatic distance, and 3) simple linear models regressing the average taxonomic and phylogenetic homogenization of the region to its native and naturalized species richness, its degree of endemism, its donor score and whether it is an island or part of the mainland. For each of the first two models, we provided their coefficient estimates and the significance of the estimates in Supplementary tables 1 and 2. For the third model, we displayed its coefficient estimates and confidence interval as points and error bars in Fig. 5.

6. I would recommend revisiting the data from island regions relative to regions with land borders as a potentially rich source of information on how biotic homogenization proceeds. Geographic isolation provides islands with a natural barrier to biotic transfer. Patterns should be reinforced in these regions, relative to regions sharing land borders and I imagine this would be a testable approach for the dataset. The Australasian (line 254) and oceanic island (line 258) results which are [presented reinforce this idea, without explicitly testing if biogeographic isolation leads to stronger homogenisation – this seems like another missed opportunity.

RESPONSE: We are not entirely sure what the reviewer means as we believe that we already did the suggested analysis. Regions with land borders are what we call mainland regions. We compared how the mean extents of taxonomic and phylogenetic homogenization of a region depended on whether the region is an island or a mainland region (i.e., a region with a land border). While islands showed a stronger taxonomic homogenization, they showed a weaker phylogenetic homogenization (Supplementary Fig. 8). It should be noted that the island effect is confounded with the area of the region, the numbers of native and naturalized species, and the proportion of endemics. Therefore, when all of those variables were included in a single analysis, islands appeared to suffer less instead of more from taxonomic homogenization than mainland regions. We discuss this in L370-375.

7. Summary: The summary overstated the need for the manuscript, attempting to present a wider knowledge gap than appears in the literature on the causes and consequences of biotic homogenisation. For instance, the statement around how ‘drastically’ the process of naturalisation has led to changes in floristic diversity is ‘still missing’ from the literature is excessive. The analyses presented do provide a wider perspective on the topic, but are by no means the first to be undertaken (and this is acknowledged by the authors in the results of the manuscript which point to islands being well-known hotspots of biotic homogenisation – one of their key findings). Work has been completed on this topic since the 1990’s and several of the papers have attracted much attention.

RESPONSE: We now toned down our statement on the novelty of our work in the summary. We do appreciate the previous studies on the homogenization topic, which have importantly contributed to our understanding of the causes and consequences of biotic homogenization. However, to the best of our knowledge, our manuscript provides the first global assessment of floristic homogenization. We would therefore

like to keep our statement on this novelty aspect.

MINOR COMMENTS

Line 130: it is not immediately clear to the reader why climatic similarity may affect the naturalisation process. This needs stronger justification.

RESPONSE: We rephrased this and added relevant citations to better justify why climatic similarity is likely to be important (L121-124).

Lines 130-134: although an expectation is given, the statement about administrative connections needs more nuance to make it a clear hypothesis. Perhaps adding the word 'increased' would help -instead of discussing the 'degree' of homogenisation?

RESPONSE: We rephrased this section to better justify the connection between administrative connections and homogenization (L126-134).

Line 137: If retained, the human development index should be correctly referred to as the 'UNDP Human Development Index (HDI)' in the Introduction.

RESPONSE: As we removed this index from our analysis, this no longer applies.

Line 176: "prerequisite" is too stronger word here – soften the language. Climate is one barrier to introduction for some species, but many species will thrive outside their climatically suitable habitat through niche shifts, or through husbandry or via genetic founder effects which resulting from the introduction of particular populations. Most simply stated, the hypothesis here is that climatic similarity predicts biotic homogenisation between region pairs.

RESPONSE: We now toned down this statement by writing: "climatic suitability is a major determinant of the establishment likelihood of alien species" (L121-122).

Line 199-200: it is not clear to the reader what the connection is between the analysis of multiple region pairs allow for testing of 'administrative relationships'. The definition of administrative relationships is not adequately developed making the connection between these two data sources seem like a non-sequitur.

RESPONSE: We agree with the reviewer that this sentence might confuse readers, and we, therefore, deleted it.

Lines 203-209. This is a much stronger way to summarises the findings, by showing the effect of naturalisation on the rate of spatial decline in taxonomic/phylogenetic similarity. This information would be better placed earlier in the discussion of the results as it is a more intuitive way of conveying the results which are about geographic distance. However, one thing to note is that the paper deals with naturalisations, not invasions and the use of this term is misleading.

RESPONSE: We thank the reviewer for this suggestion. We moved the text to an earlier place (L176-184) before the analysis of how the geographic distance, climatic distance, and administrative relation between region pairs affect the change in their floristic similarity by naturalized species and comparison of homogenization across different climatic zones.

We politely disagree with the reviewer that our paper does not deal with invasions because naturalization is one of the stages of the invasion process. Nevertheless, as

many readers might hold the opinion that invasion refers to invasive species only, we decided to replace 'plant invasion' by 'naturalized plants'.

The writing style and expression requires attention throughout. For instance, sentences are often long and as a result hard to comprehend. Examples include lines 112-116, or lines 134-140 which is six(!) lines long and full of different concepts. The expression is often quite 'loose' in that bold statements are made without sufficient nuance (e.g. line 176 about climatic suitability) or sentences are poorly constructed (e.g. line 178 ends with 'will be'). I understand that Nature format requires brevity, but this should not come at the expense of clarity.

RESPONSE: We checked the manuscript, restructured and rephrased many sections, and shortened or broke up the unnecessarily long sentences into shorter ones.

Line 245: eucalyptus species, not trees – the results are not at an individual level, but at a species level.

RESPONSE: We replaced "trees" with "species" (L310).

Line 270: I wanted a little more information from this finding about small island having high phylogenetic similarity to native floras of other regions. Seems to me that this result will be due to the processes involved in island formation, which may cut off areas previously connected to mainland regions (e.g. New Caledonia and Australia).

RESPONSE: Many of the small islands are oceanic islands, which emerged from the ocean by volcanic activities or by uplifting of the ocean floor due to tectonic plate movement. This means that native oceanic island floras are the result of colonization events, particularly from mainland areas, and subsequent diversification. Therefore, island floras are phylogenetic subsets of other floras that served as donors. We now expanded the discussion on this topic (L359-369).

It would be nice to see fewer analyses and more depth in the interpretation of what is offered. For instance, here's a significant amount of inference being made here about how colonial histories may shape trading of species. While I do not disagree that this is a potential pathway for floristic exchange, I would have liked to have seen a stronger suite of predictors to really tease out this result.

RESPONSE: In response to one of the other comments of this reviewer, we removed the analysis of the effects of anthropogenic factors on the mean homogenization of regions. This allows us to focus more strongly on the patterns and causes of the change in the floristic similarity between regions. To test the role of colonial histories in shaping species exchange, a suite of predictors such as trade and human migration between regions would be more direct. We have intensively looked for such data, but unfortunately, such data are available for very few regions and usually only for regions that correspond to countries or states on the mainland. More importantly, these data are only available for recent decades, whereas many of the alien species may have established already centuries ago. We, therefore, used colonial histories and administrative ties as general indicators of the past and current connections between regions. People are more likely to migrate between regions with current or past administrative ties, and trade agreements between the different colonies of an empire increased the connectivity of those regions. We now explain this in more detail (L126-134).

Reviewer #2 (Remarks to the Author):

The study by Yang et al. provides a fascinating and important global-scale analysis of the impact of plant naturalizations to floristic homogenization – that is, the loss of uniqueness in composition among evolutionarily and historically distinct biogeographic floras. At first, I was skeptical of the claim in the Abstract that information on the contributions of non-native plants to biotic homogenization is “still missing,” as there are many studies on biotic homogenization in the context of invasions and the concept is far from understudied. However, as I read on, the analyses presented are indeed very unique, which have similarly been done at smaller scales but only speculated at the global scale. The authors find: 1) Homogenization is occurring at global scale, with naturalizations overwhelmingly increasing taxonomic and phylogenetic similarity between distant floras – the expected decay in similarity with distance is decreasing due to anthropogenic mixing of floras; 2) Homogenization is more severe between broadly climatically similar regions; 3) Homogenization is greater between regions that were historically or currently connected through administrative relationships (eg. colonialism). There are many interesting sub findings within those conclusions as well, including interesting island-mainland differences. Though the results are not entirely surprising (though some aspects are unexpected), this study provides a quantification of the phenomenon of anthropogenic mixing of regional floras. It has the potential to be highly cited and to spur additional research on the multidisciplinary study of causes behind the patterns, and their past, present, and future impacts. Isolated floras are no longer isolated, and the analyses in this paper leverage impressive datasets on both native and naturalized floras to quantify the impacts of floristic interchange in the Anthropocene.

RESPONSE: We thank Dr. Heberling for his positive and insightful comments on our manuscript. Please note that we numbered the major comments of this reviewer for ease of reference.

1. I am unclear on the impact of the region size on the results. I think this concern is minor, as I suspect this is my confusion. The authors combine many checklists of native and naturalized species at various scales to create contrasts among 658 regions. However, as stated in Methods (L315), the regions represented in the GloNAF database are hugely variable in area, ranging from 0.03 sq. km to nearly 7 million sq. km. However, I am unsure on the criteria for merging these regions in the 658 regions used in the analysis. Were region defined based on biological or practical criteria? What is the range of area in these regions? Are the metrics used in the contrasts sensitive to size of regions? I would expect that both the geographic and climatic distances used in analyses to be particularly sensitive to region size disparities between the contrasts, given that the distances are calculated based on geographic and climatic centroids (L373-390). Further, how were regions “...judged to include more than 50% of the naturalized taxa for that region” (L329)? The authors state Simpson similarity metrics used are robust to species richness, but I am unclear if results in this context are also insensitive to region size. Maybe it does not matter. Ideally contrasts of regional floras could be defined based on some a priori biogeographic (such as Takhtajan’s floristic regions or similar) rather than more arbitrarily defined regions.

RESPONSE: There is indeed a large variation in the size of the regions in our analyses. The regions are mostly administrative units, corresponding to countries,

provinces, and states (e.g., the different USA states), as the checklists of native and alien floras are usually only available for administrative regions. Therefore, the large variation in size of the regions is unavoidable, and it is not possible to use biogeographically defined regions. So, when regions were merged, this was done for practical reasons. To address how the size of the regions relates to homogenization, we analysed how the mean extent of homogenization of a region relates to its size, as well as to several other variables (native species richness, degree of endemism etc.), and we report these results in Fig. 5.

We agree with the reviewer that geographic and climatic distances might be sensitive to region size. In response, we now also conducted analyses using other measures of geographic and climatic distances between regions. For geographic distance, we used the shortest direct distance between the borders of the regions as an alternative to the distance between geographic centroids. For climatic distance, we calculated an alternative measure that considers the climatic distances between all grid cells of a pair of regions. First, for each grid cell of a region, we calculated the differences in climatic conditions with all grid cells of the other region. From these climatic distances, we then calculated the average of the differences as the new measure of climatic distance between the regions. This was done for the overall climatic similarity, as well as for the similarities in climatic PC1 (mainly related to variation in temperature) and PC2 (mainly related to variation in precipitation). Overall, the results were very similar to the ones we had already. As other reviewers are also concerned about the contribution of distance in specific climatic variables to the change in similarity caused by naturalized plants, we, therefore, present the new results using PC1 and PC2 in the Supplements (Supplemental Fig. 6-7).

The judgment that a species list includes more than 50% of the naturalized taxa for this region was in most cases made by the authors of the GloNAF database (in a few cases, it was based on the judgment of the people that had compiled the original data source). Admittedly, this is a coarse estimation, as it is impossible to know the exact percentage. However, the main aim was to exclude regions for which the data are obviously very poor. We now explain this more thoroughly in the manuscript (L454-458).

2. I applaud the authors for beautiful and easy to understand figures. There is a lot to take in!

RESPONSE: We thank the reviewer for this compliment.

3. Global invasion patterns are notably asymmetric with some regions more likely to be donors than recipients, as has been well shown regionally in several studies and globally by a subset of the current authors (van Kleunen et al. 2015 Nature). This notion of asymmetric species exchanges could be important to include in the introduction or conclusion in the context of homogenization. Further, I am curious: are “predominantly donor” regions less likely to be homogenization hotspots?

RESPONSE: This is an interesting point, and we now mention the asymmetry of donor and recipient regions in the Discussion (L375-383). However, we do not think that predominantly donor regions should be less likely to be homogenization hotspots. On the contrary, if a region has donated many species to many other regions, this would mean that its flora now shares more species with the other regions, which should

contribute to homogenization. We now explicitly tested whether there was a correlation between the mean extent of homogenization and the donor score of the regions (Fig. 5). For each native species of a region, we calculated the average number of the non-native regions in which it is naturalized.

4. I assume the GLONAF dataset does not comprehensively capture "invasive" species from "naturalized" species. It may be interesting to speculate on the role invasive species have on realized homogenizations, as many naturalized species are limited in their extent and/or abundance.

RESPONSE: Indeed, as the data sources used for GloNAF either do not indicate which naturalized species are invasive or if they did, did not all use the same definition of 'invasive', we decided not to use these data here. As mentioned in our response to comment 3 of Reviewer 1, we do not have data on the abundance of the species. However, we now mention in the Discussion that the abundance of naturalized species may theoretically also contribute to changes in homogenization. We also mention that we expect that invasive species, which usually have larger ranges and thus are naturalized in more regions, will contribute more strongly to homogenization than the ones that are only naturalized (L407-415). Please, also see our response to the last point of comment 3ii of Reviewer 3.

A few minor comments:

L146: An equation here (or in Methods) to show the components and calculation of the similarity metrics would be helpful. A statement that greater positive values indicate homogenization, while more negative values indicate heterogenization.

RESPONSE: We added the equation of Simpson similarity in the Methods (L481), and we now mention that positive values indicate an increase in similarity (homogenization), while negative values indicate a decrease in similarity (L492-493).

L188: Please give test statistic with p value.

RESPONSE: In the comparison of homogenization across different climatic zones (Fig. 4), we used paired Wilcoxon test. As all comparisons ($C_6^2 = 15$) are significant, we do not give the value of the test statistic for each comparison. Instead, we mention that all of the pairwise comparisons between groups are significant (L220).

We now also provided additional information on the significance of the coefficient estimates of the main regression models. Please also see our response to comment 5 of reviewer 1 for more details.

L226: "...contain signals of colonial history dating back at least 500 years." I think some elaboration is needed, as I was unclear where this 500 years originated. The oldest administrative connection in dataset?

RESPONSE: In response to a comment from Reviewer 3, we decided to remove the part about 500 years (L342).

L264: The finding of heterogenization on some islands was unexpected and interesting. The reasoning of higher phylogenetic similarities with mainland floras makes sense.

RESPONSE: In response to the comment of Reviewer 1 to line 270, we expanded the

Discussion on this interesting result (L359-369). However, we are pleased to read that Dr. Heberling thinks that our explanation makes sense.

Figure 2: Please add sample sizes for boxplots in the caption and/or figure and/or supplement.

RESPONSE: We now added sample sizes to all figures.

Figure 3: Please define meaning of “dependency” in caption.

RESPONSE: We now added the definition of dependency in the caption of Fig. 3 (L698-701).

Figure 4: Perhaps add that negative values indicate heterogenization in the caption for clarity.

RESPONSE: We now added that negative values indicate a decrease in similarity (L722). Note that, on request of Reviewer 1, we no longer use the term ‘heterogenization.’

The code seems very well annotated and accessible, which was no easy task given the scope of the analyses and datasets. Only a gentle suggestion - it could also perhaps be permanently archived to allow static dataset and versioning.

RESPONSE: We thank Dr. Heberling for the compliment on our code. We now archived both the code and dataset in figshare, and made the link public. Please, see, Data and Code availability.

A lingering thought, perhaps best for future work:

It would be fascinating to also analyze functional homogenization – how is trait similarities/diversities changing? Does phylogenetic homogenization match functional homogenization? GIFT includes traits, so perhaps this was already part of the research team’s agenda.

RESPONSE: We thank Dr. Heberling for these suggestions. We would indeed be very interested in exploring functional homogenization and how it relates to phylogenetic homogenization. However, although GIFT includes traits, for most species the trait data is very incomplete. Hopefully, the increasing availability of traits will allow us to do such an interesting analysis in the future.

Mason Heberling
Carnegie Museum of Natural History

Reviewer #3 (Remarks to the Author):

SUMMARY

This paper by Qiang Yang and colleagues aims at quantifying both: the native floristic similarity between regions of the world, and whether this similarity is affected by the presence of naturalized alien species. They carry their analysis at a global spatial scale, on 60% of all native angiosperm species and >10’000 naturalized alien angiosperm species. The effects of alien species on (taxo and phylo) similarities between regions is then analysed in regards to several factors including geographic distances, climatic differences, administrative histories, levels of human developments, and biodiversity intactness. They found that the presence of naturalized

alien plants generally increases both taxonomic and phylogenetic similarities between regions.

WRITING & PRESENTATION

The manuscript has been carefully written and I enjoyed reading it. It is also well organized and the figures are nice.

RESPONSE: We thank the reviewer for the positive comments on our manuscript.

CONTEXT & ORIGINALITY

I agree with the authors that an analysis of (taxonomic and phylogenetic) similarities at such large spatial and taxonomic scales is novel for plants species and potentially interesting. However, under its current form, the manuscript could not convince me about how the results are important for our understanding of the threat posed by invasive species, and whether their methodological choices are the most relevant ones to understand invasion-driven biotic homogenization (see my major comments below).

RESPONSE: We thank the reviewer for putting forward these insightful comments. Before making specific responses to these comments (see below), we first would like to point out that our manuscript is not about the threat posed by invasive species. Our paper is about naturalized species, i.e., alien species that have established self-sustaining populations in the wild and thus have become persistent members of the regional flora. Some of these naturalized species are considered invasive in that they impact native plants, ecosystem functions, or humans. However, although such invasive species might contribute more strongly to global floristic homogenization (see our response to comment 4 of Reviewer 2), we here consider all naturalized species irrespective of their invasiveness status. We focus on the change in regional species composition and not on the negative impacts of invasive species.

GENERAL COMMENTS

I have 3 general concerns that I think should be answered, they include: (1) the need to clarify the general aims and hypotheses of the study, (2) the need to clarify the link between invasion-driven homogenization to the other tested factors, as well as the methodological choices associated, and (3) a series of methodological questions. I detail them below.

(1) The need to clarify the general aims and hypotheses of the study.

I think it is important to explain the reader: (i) why invasion-driven homogenization is a threat to biodiversity and not a good thing (as this kind of homogenization is about species gains, and not species losses)? And (ii) why one should care about phylogenetic uniqueness of a flora?

(1.i) L. 80 (3rd sentence in the intro): “Importantly, [alien species] also change patterns of biotic uniqueness or □distinctiveness of those regions compared to others, with potential ecological and □evolutionary consequences (Ref 4, 16)”. This is a key sentence as it sets the rationale and motivation for the whole study: why it is important to understand invasion driven homogenization. However, the authors never explain why it is “important” and what are those “potential ecological and □evolutionary consequences”. This lack of explanation questions the actual relevance of such analyses. In fact, one can genuinely wonder why adding more species to a region has important eco and evo consequences, and whether these consequences would be

harmful to the ecosystem. I guess that one part of the answer is that naturalized species may harm in some way native species, and thus can lead to local biodiversity reduction (argument proposed in their ref 16), but this is not really what is investigated here (as there is no data on native species range reduction for example).

(1.ii) Why should people care about phylogenetic uniqueness of a flora? The phylogenetic aspect of uniqueness is mentioned L.109, but without an explanation about why it is interesting to understand this aspect. What is the argument? What means “unique evolutionary history” (I understand the words, but not the ecological implications), and why should we account for it in conservation (ie. is it a proxy for functional uniqueness? Or is there something else?)?

Additionally, some “expectations” for the different analyses of this work are listed in the introduction (e.g. L.134-140), but the rationale behind these expectations are not explained to the reader. Please add these explanations, and indicate whether these are novel speculations or results already found in the literature.

Don't get me wrong, I am sure the authors have plenty of good scientific motivation for their work, but it is essential that they make these motivations explicit to the reader so that one can take the measure of the value of this work and its implications for our understanding of invasion and protection of biodiversity.

RESPONSE: We have now restructured our manuscript and rewritten large parts of the Introduction to better clarify the general aims and hypotheses. We would like to point out, though, that our analysis is not only relevant because invasions might pose a threat. Understanding the distribution of biodiversity is a primary goal of ecology and biogeography; documenting changes in species composition, therefore, has fundamental value, irrespective of whether the changes are 'good' or 'bad' from a given perspective.

(2) The need to clarify the link between invasion-driven homogenization to the other factors, as well as the methodological choices associated:

I did not really understand the usefulness of linking invasion-driven homogenization to the other factors, please clarify your motivations (or consider removing these analyses).

RESPONSE: In response to this comment and comment 2 of Reviewer 1, we decided to remove the analyses linking the mean extent of homogenization of regions to anthropogenic factors. However, with regard to the pairwise similarity of regional floras, we wanted to go beyond just quantifying the degree of homogenization by also trying to explain why the floras of certain regions have become more similar to each other than those of other regions. We expected that the extent of homogenization would increase with geographical distance because nearby regions already have naturally similar floras, and humans helped the species to overcome the barrier of long-distance dispersal. We also expected that, after correcting for geographic distance, this would be stronger for regions that are more likely to have had intensive transport of goods and people between them, such as the regions that had or have administrative relationships. Furthermore, we expected that this would be stronger for environmentally, and particularly climatically, similar regions, even if these regions are in different hemispheres, as the alien species might be pre-adapted to those conditions. We now explain this more clearly in the Introduction.

Here are a few examples of my lack of understanding:

(2.i) Why is it interesting and important to link homogenization to geographic

distances? What I understood is that we learn that invasive species are less sensitive to environmental barriers than native species. I do not really understand the novelty of this finding (it is somewhat in the definition of invasion). If there are other conclusions that are drawn from this analysis, please make them more explicit. Otherwise, I suggest removing it.

RESPONSE: We now explain that distance decay of biotic similarity is a well-documented biogeographic pattern. And thus that it is not surprising that we find this pattern in our dataset. However, the rate of decay in similarity can vary depending on the circumstances. A low rate of decay (slope) indicates slow species turnover and generally a high similarity. A high rate of decay indicates high species turnover and lower similarity (less homogenous). The difference in the distance decay relationship with and without alien species, therefore, provides a means of quantifying biotic homogenization at the global scale. Moreover, one needs to account for this geographic effect (i.e., distance decay) in order to test the roles of potential additional drivers of homogenization (such as climate and colonial history).

(2.ii) L. 63-65 (abstract): “floristic homogenization was greater between regions with current or past administrative relationships than for regions without, indicating that trade, transport and colonial history facilitate floristic exchange”. I did not see what was new in this finding: both trade and transport are the means by which invasions occur, by definition. The one new thing for me here is the colonial history part, but then it is not really a driver of invasion per se, rather an indicator of the intensity of transport and trade, right? Please clarify this point. Additionally, if the idea is to show that colonial history has favored invasion, most likely via increased transport and trade, wouldn't it be better to use a more direct test of this than going through homogenization measures? (e.g. testing whether species probability to invade a region depends on whether it is native to another region of the same colonial empire; or use SEMs with all info together: colonial history, trade, transport, invasion; or any relevant alternative).

*RESPONSE: We agree that it is not a surprising result that floristic homogenization was greater between regions with current or past administrative relationships. Nevertheless, we are not aware of any previous study that has actually quantitatively assessed and analysed this before. Of course, there are studies that have shown that trade and transport are related to the number of naturalized species (e.g., Seebens et al. 2015, *Global Change Biology* 21:4128-4140), but they did not look at species composition and used only trade or transport data for the last couple of decades. The reason is that such trade and transport data do not exist for earlier times, at least not at the global scale. As many naturalizations happened already centuries ago, they obviously cannot be explained by recent trade and transport. Moreover, data on trade and transport are usually only available at the country level. This would mean that we would have to discard many regions, and particularly islands, from our analysis. Therefore, we chose instead to use the past and current administrative relationships as an integral proxy of trade and transport between regions. It is known that trade agreements between the different colonies of an empire intensified trade among those regions (Mitchener and Weidenmier 2008, *The Economic Journal* 118: 1805-1834). It is also known that there was an active exchange of species among gardens within the colonial empires (e.g., McCracken 1997, *Gardens of Empire*). We now mention this in L126-134.*

The reviewer suggests that there are also other ways to test whether being part of the same colonial empire affects the naturalization probability of a species within the empire. We agree that this can be tested in different ways, but as we focus in this study on homogenization, we wanted to analyse it in this context instead of switching to a species perspective.

Similar concerns can be applied to the other factors, please take the time to clarify the text in this regard. [e.g. L.276-278: “Taxonomic homogenization decreased with biodiversity intactness, indicating that regions in which there is less loss of native biodiversity have also suffered less from plant invasions.” I do not understand why you need to go through analyses of beta-div to test such hypothesis. Same reasoning works for the following (L. 281-282) about the link between invasions and human populations.]

RESPONSE: As mentioned above, we removed the analyses of how the mean extent of homogenization changes with biodiversity intactness and other anthropogenic factors. Therefore, this criticism no longer applies.

(3) Methodological questions and concerns:

(3.i) Do you need null models to quantify significant alien species effects?

It seems to me that there could be a mechanical effect of species richness on the change of similarity when considering only native species or native+invasive species. Isn't it correct that strong homogenization by alien species is much easier for communities with few native species than for species rich communities, as invaders can quickly become a large proportion of the dissimilarity between regions? If so, could this effect be controlled by the use of a null model, testing “are the observed effects of alien species on region similarities, greater or smaller than those expected in two regions of the same taxo richness but containing a random selection of species?” (or any other more relevant null model if you think of a better one). The random expectations could be calculated from random assemblages constructed using the ‘independent swap’ algorithm that randomizes species co-occurrence but keeps constant the sample richness per community and frequency of occurrence of each species across all communities (Gotelli & Entsminger, 2003). Doing so would also help quantifying whether the alien species effect are “significant” or not. Similarly, regarding the alien species effects on phylogenetic uniqueness, a null model could be used to quantify whether the invasive species are creating significantly more phylo-homogenization than expected by random addition of species (e.g. by randomization of the tips of the phylogeny?).

RESPONSE: The reviewer asks whether we need a null model. We do not think there is a need to justify the significance of global homogenization of regional floras through randomization. Firstly, one of the strengths of this analysis is that we are documenting the observable outcome of species naturalizations on species-composition patterns. Our results quantify how much floristic similarities between regions have changed with the addition of naturalized alien species, and we do not need randomizations to confirm whether or not this is the case. The randomization suggested by the reviewer (shuffling native species while maintaining species richness) would tell us whether or not the degree of observed homogenization is more or less than expected under a scenario where some of the structure of native floras has been removed. Although this comparison may reveal something about the similarity of native communities, it would not provide a meaningful baseline expectation for the effects of naturalization. Secondly, homogenization is a function of

both the similarity between native floras and the similarity between floras that include both native and naturalized species. It is true that the same number of naturalized species might cause a greater change between floras of lower native richness (because they comprise a higher proportion of the total number of species). However, the change in similarity will not be biased towards homogenization because it could equally cause a great change towards dissimilarity. In addition, the number of naturalized species is positively correlated with the number of native species (Pyšek et al. 2017, Preslia 89:203-274; therefore, the species effect on the relative change in similarity could not be logically expected. Regardless, we have now analysed how the mean extent of homogenization of a region relates to the number of native species, and we included this in Fig. 5.

(3.ii) Also, I think that it would be important to understand a little better what drives the differences in similarities across region pairs. Are they driven by their level of endemism of native species and phylogenetic distinctiveness? Are there some effects of the region area on the results (cf. Barton et al 2013 GEB)? And is it linked to local native richness, alien richness, or alien:native ratio? Are most of the homogenization patterns driven by the most widespread invaders, while the heterogenization patterns are rather driven by the “narrow” range invaders (and if so is there a sampling bias toward widespread invaders in the dataset, with which consequences for the results)?
RESPONSE: We thank the reviewer for putting forward these interesting questions, to which we respond below.

- “Are they [differences in similarities] driven by their level of endemism of native species and phylogenetic distinctiveness?”
RESPONSE: We are not entirely sure why the reviewer asks this because endemic and phylogenetically distinct species should by definition contribute more to dissimilarity than non-endemic and phylogenetically similar species. Nevertheless, we now analysed how the mean extent of homogenization relates to the proportion of species that is endemic to a given region (Fig. 5).
- “Are there some effects of the region area on the results?”
RESPONSE: We now tested whether the mean extent of homogenization depends on the region area (Fig. 5). Moreover, in the analyses of the changes in pairwise floristic similarity, we now also used some alternative measures of geographic distance and climatic dissimilarity that should be less sensitive to the region area (see our response to comment 1 of Reviewer 2).
- “And is it linked to local native richness, alien richness, or alien:native ratio?”
It is a strange question, as these are the components that directly used in our calculation of homogenization.
RESPONSE: We now tested how the mean extent of homogenization of a region depends on the native richness and naturalized alien richness (Fig. 5). We did not include alien:native ratio separately because in the multiple regression framework we used, the effect of alien richness is automatically corrected for native richness and vice versa.
- “Are most of the homogenization patterns driven by the most widespread invaders, while the heterogenization patterns are rather driven by the “narrow” range invaders (and if so is there a sampling bias toward widespread invaders in the dataset, with which consequences for the results?)?”

RESPONSE: To test whether most homogenization patterns were driven by the widespread invaders, we first removed the 10% most widely naturalized species, and then we removed the 10% least widely naturalized species instead. We compared the homogenization pattern after the two removals to the originally observed homogenization pattern without removal of naturalized species and present the results in the supplementary information (Supplementary Fig. 10). These analyses confirm that the more widely naturalized species contribute more to homogenization, which we now mention in the Discussion (L268-276).

GloNAF is not a compilation of distribution data for individual species but a compilation of regional lists of naturalized species. It could be that species that are more widespread in a region are more likely to be included on the list of naturalized species for that region. However, this does not mean that the species will be naturalized in more GloNAF regions. Therefore, we believe that there is no sampling bias toward widespread invaders in the dataset.

(3.iii) Finally, I think that it is necessary to discuss the results in regard to the potential weaknesses of the dataset. Typically, are the missing species (40% of angiosperms missing) randomly spread in the phylogeny, and where are the alien species in these “data holes”? This would help assessing whether the phylo-similarity and homogenization can be biased toward over- or under-estimations. Also, some homogenisation cold spots are in asia, central America & Amazonia, central Africa (Fig. 4); are these influenced by the data quality there?

RESPONSE: We now added a figure to present the phylogeny of the global angiosperm flora (Supplementary Fig. 2). We marked the position of the missing species with bars on the phylogeny. Although there is a significant phylogenetic signal, the missing species are found to be distributed around the entire phylogeny.

We do not think that the homogenization cold spots are a consequence of poor data quality for those regions but are more likely to reflect that in those regions, the anthropogenic pressures are still lower. However, as we were asked by Reviewer 1 to remove the analyses on anthropogenic pressures from the manuscript, we do not mention these explanations in the manuscript.

SPECIFIC COMMENTS

References: Within the first 15 references I counted 10 references including at least one co-author of this manuscript. I understand that many of the authors have been working on invasions for a long time now and have published a lot of good work on related research areas. However, I believe it would bring a little more credit to the generality of this work if they would cite a little less of their own papers.

RESPONSE: We did not intend to only cite our own work, but as many of the global scale analyses are based on the GloNAF database, it automatically happened. However, we now deleted several less important references to papers published by the coauthors, and we added references to papers by others.

L.58: “significant” this is misleading as it somehow suggests $p\text{-value} < 0.05$ while there was no test here.

RESPONSE: We agree and changed the word “significant” to “strong” (L58).

L.100. “the size ‘OF’ the different sets..” typo?

RESPONSE: We added the word “of” (L96).

L.167: I do not understand the argument here, if species are alien to both regions then they can only homogenize them (as explained in the introduction), no? why do you here expect heterogenization?

RESPONSE: We thank the reviewer for this comment. It is important to realize here that if a species is alien to a region, this does not mean that it is naturalized in that region. There are two situations in which one species alien to both regions could change the similarity between pairwise regions. When the alien species naturalizes in both regions, it increases the similarity of the regions. However, when the alien species is naturalized in only one of the two regions, it could decrease the similarity between them. We altered this sentence to further clarify our point (L93-95).

Moreover, in response to comment 1 of Reviewer 1, we also made a new version of Supplementary Fig. 1, which we now include in the main manuscript. Moreover, we added an extended version of this figure with examples in the supplementary information (Supplementary Fig. 1).

L. 172-174: “So, the decrease in floristic similarity of two regions is not solely □driven by the exchange of species that are native to one of the two regions, but also by □alien species from elsewhere that have invaded both regions.” This is not really a finding of your study, but rather a description of how you measure homogenization. Or did I misunderstand the argument here?

RESPONSE: We agree with the reviewer that this is indeed not a finding of our study. We removed this sentence to avoid confusion.

L. 213-217: “... the degree of taxonomic homogenization increased more rapidly with □geographic distance for region pairs belonging to the same country than for other □region pairs (Fig. 3c, Supplementary Figs. 3&5d). This most likely reflects facilitated □spread of an alien species within a country after it has become established in one of □the subregions.” As mentioned in the general comments, if that is the question, then why don’t you just analyze the probability of naturalization as a function of administrative borders, or colonization history?

RESPONSE: Please, see our response to comment 2.ii of this reviewer.

L.226-227: “floristic homogenization contains signals of colonial history dating back at least 500 years”. I don’t see how you can say that the signal dates from 500 years here. Maybe the signal you found is just driven by what happened the last 100 years? Either remove this statement or test explicitly the dating (e.g. use different colonial subsets of different time periods).

RESPONSE: We agree with the reviewer that we have no evidence of the exact date of the signal. However, what we wanted to point out is that we considered colonial relationships during the last 500 years statement. We now deleted the part of the sentence about 500 years (L342).

L.240-243: sentence unclear, please breakdown or clarify. (this one paragraph reads less well than the others, with a lot of terms like “increase” “decrease” “similar” “dissimilar” “distance” “distant” “nearby”, you may want to simplify it).

RESPONSE: We now rephrased the sentence to increase its readability (L304-L310). We also revised the paragraph to reduce the usage of synonymous terms to increase the readability (L303-312).

L.270-273: I did not understand this argument, please clarify.

RESPONSE: We rephrased this sentence to better clarify what we meant (L359-361).

L. 349: I calculate 215,824 pairwise combinations of different regions ((658x658)-658=215,824), and not 216,153 as indicated.

*RESPONSE: We calculated the number of unique combinations of different regions: $(658*658 - 658)/2 = 216,153$*

L368: please add the number of alien species that are part of these missing species in the phylogeny (to clarify that the aliens were not particularly badly resolved).

RESPONSE: We now added the number of missing alien species (L501).

L.435: please add a ref for this null distribution of coeff approach?

RESPONSE: We now added the relevant references. Lichstein, J. W. (2007) Multiple regression on distance matrices: a multivariate spatial analysis tool. Plant Ecol. 188, 117-131 (L577).

- Please add the formula of the Simpson similarity index used (in the sup mat if your are limited by space in the main text), and a sentence to explain why you chose this single index to describe similarities across regions (e.g. and not a pair of turnover & nestedness components)? As it is written in the method section now it seems that this index choice is due its insensitivity to species richness differences across regions, but a null model approach could be used to solve this issue, no? is there another reason?

RESPONSE: We now added the formula of the Simpson similarity index in the Methods (L481). Yes, we indeed choose the Simpson similarity index as it is insensitive to the species richness differences across regions. We now also calculated the Sørensen similarity index and present plots of the taxonomic and phylogenetic similarity with naturalized aliens vs the similarities without naturalized aliens in the supplementary information (Supplementary Fig. 13). These results are very similar to the ones based on Simpson similarity, which is not surprising given that they are related.

Please, see our responses to comment 3.i for why we did not use the null model approach.

- Maybe worth adding a sentence in the methods or discussion explaining what are the implications of using artificial (here political units such as states, provinces, counties) rather than natural biogeographical regions when quantifying biotic homogenization? (even though I understand that the data per biogeo region is not available)

RESPONSE: We added the following sentence in the Discussion to address the limitation of using administrative regions: "Another limitation is that because species lists are mainly available for administrative regions, and not for biogeographic regions. As a consequence, the units vary largely in size, environmental heterogeneity and native species richness."(L402-404).

Reviewers' Comments:

Reviewer #2:

Remarks to the Author:

This is a review of revised version of a study by Yang et al., which I reviewed earlier this year. The authors succinctly and clearly addressed my concerns and questions in manuscript and well justified response letter, including the addition of a new analyses and added clarifications. I have no additional concerns or comments. Their study is impressive and well presented-- no small feat to assemble and analyze this complex dataset. There are certainly caveats and the authors address directly in the manuscript. The findings are clearly supported and well described. This study is a solid and important contribution to the literature on global patterns of plant naturalization, and likely influence future study on the topic.

Mason Heberling

Carnegie Museum of Natural History

Reviewer #3:

Remarks to the Author:

This is the second time I have evaluated this manuscript (Reviewer #3 in the last round).

GENERAL COMMENTS ON THE REBUTTAL

The authors have provided substantial amount of work for this new version of the manuscript, and I appreciate the improvements. Although I do not fully agree with all the responses provided in the rebuttal, I acknowledge that my remaining disagreements do not affect the main results of the study. Please find below my final list of concerns that I believe deserve consideration prior to publication.

REMAINING CONCERNS

1. The discussion of the abundance of naturalized species and its effect on homogenisation of the flora (Question raised by Rev #1):

This point was raised by Rev #1, and I think it is fair point that could be better handled in the discussion section. In fact, even though abundance values may have little effect on similarity index values, it still raises the question of what it means for "the homogeneisation of the flora" to have naturalized species with either (i) only few populations restricted to a small area, or (ii) a broad coverage of study region. Maybe the flora is not so homogenized after all if there are only a handful of naturalized pops, right? I would be more moderate in discussing this point at L. 407-415 (and not mention the impacts of "invasive" species there since you are working on naturalized species).

2. Remove results about tropical vs non-tropical regions:

As already discussed by Rev #1, I think that the tropical vs non-tropical distinction brings more confusion than it helps (as it can not separate the effects of geographic distances from climatic distances or biogeographic histories; e.g. L. 285-289). In fact, the rationale for these comparisons across broad regions is not even presented in the introduction.

In their rebuttal, the authors make the following argument for retaining the tropical-non-tropical results: "We therefore strongly argue that the previous Fig. 2 and associated analyses provide a valuable complement to the more detailed climate analyses by intuitively showing that e.g., homogenization is most pronounced for distant regions in similar climatic zones."

In the manuscript, they explain that homogenization is (i) highest between extra-tropical regions in different hemispheres, and that (ii) within the same hemisphere homogenization between extra-tropical regions in the southern hemisphere is greater than in the northern hemisphere.

But can those results not be due solely to geographic distance effects: (i) extra-tropical regions of the different hemispheres are furthest apart (relative to all other types of pairwise comparisons), and (ii) extra-tropical regions in the northern hemisphere are on average less far apart than the regions of the southern hemisphere (because there is so much ocean in the south)?

Overall, wouldn't it be simpler (and more logical) to draw conclusions, about the interaction between geographic and climatic distances, from Fig. 3 c and f, which show it via the colors

associated with the different climatic bins?

[Also, in L. 224-225, the authors describe that “phylogenetic homogenization ^[1]_{SEP} was lowest for regions that are both in the tropics (Fig. 4)”, but this point is not explained or interpreted afterwards.]

3. Method points:

Why did you use GLMs for the simple models (L. 565-566), and then MRM for the models with multiple predictors (L. 571-574)? I probably missed a point here, but if not, this lacks consistency in approach and should be homogenized.

Also, I just realize now that the Fig. 3c and the Fig. S5d are not “qualitatively” aligned as argued in the L. 583-585. The colour gradient shows opposite directions for the “dependence” group of administrative relationships. (same issue for the “same country” class in the phylogenetic measures Fig. 3f vs Fig S5h). What are the implications of these differences on the results?

SPECIFIC COMMENTS:

L. 70: I guess you mean “unique” instead or “similar”?

L. 401-402: “this ^[1]_{SEP} might have caused some biases, but this is unlikely to change the overall finding of a ^[1]_{SEP} loss of regional floristic uniqueness”. Please add an explanation about which biases you expect, and why you think it is unlikely to change your results.

L. 391-394: the sentence is quite difficult to understand, consider reformulation to facilitate reading.

Dear editor,

We thank you for your invitation to revise our manuscript according to the constructive comments of the reviewers. Below you will find our point-by-point responses (in blue) to the reviewers' comments.

We hope that our revised manuscript will now be acceptable for publication in Nature Communications.

Sincerely,

Qiang Yang (on behalf of all authors)

REVIEWER COMMENTS

Reviewer #2 (Remarks to the Author):

This is a review of revised version of a study by Yang et al., which I reviewed earlier this year. The authors succinctly and clearly addressed my concerns and questions in manuscript and well justified response letter, including the addition of a new analyses and added clarifications. I have no additional concerns or comments. Their study is impressive and well presented-- no small feat to assemble and analyze this complex dataset. There are certainly caveats and the authors address directly in the manuscript. The findings are clearly supported and well described. This study is a solid and important contribution to the literature on global patterns of plant naturalization, and likely influence future study on the topic.

Mason Heberling
Carnegie Museum of Natural History

Response: We thank Dr Heberling for the positive feedback and evaluation of our manuscript.

Reviewer #3 (Remarks to the Author):

This is the second time I have evaluated this manuscript (Reviewer #3 in the last round).

GENERAL COMMENTS ON THE REBUTTAL

The authors have provided substantial amount of work for this new version of the manuscript, and I appreciate the improvements. Although I do not fully agree with all the responses provided in the rebuttal, I acknowledge that my remaining disagreements do not affect the main results of the study. Please find below my final list of concerns that I believe deserve consideration prior to publication.

Response: We thank the reviewer for the constructive comments. We also thank the reviewer for considering our responses to the comments of Reviewer #1.

REMAINING CONCERNS

1. The discussion of the abundance of naturalized species and its effect on homogenisation of the flora (Question raised by Rev #1):

This point was raised by Rev #1, and I think it is fair point that could be better handled in the discussion section. In fact, even though abundance values may have little effect on similarity index values, it still raises the question of what it means for “the homogenisation of the flora” to have naturalized species with either (i) only few populations restricted to a small area, or (ii) a broad coverage of study region. Maybe the flora is not so homogeneized after all if there are only a handful of naturalized pops, right? I would be more moderate in discussing this point at L. 407-415 (and not mention the impacts of “invasive” species there since you are working on naturalized species).

Response: Our study focused on the homogenization of regional floras, which are merely checklists of species for a country, state or province. While an alien species that has only a few populations restricted to a small area in a region contributes to a change in the regional floristic similarity, we agree with the reviewer that this alien species might contribute less to homogenization at smaller local spatial scales. We now write more clearly that we refer to regional floristic homogenization, and we now mention that regional floristic homogenization does not necessarily mean that local communities within those regions show the same degree of homogenization. We now also clarify that invasive species should contribute most to homogenization, if one could consider abundances (lines 397-409):

“Theoretically, however, changes in species’ occurrences throughout a region or their local abundances could also influence regional floristic homogenization. If one could account for abundance, the naturalized species that have become widespread and invasive (i.e. abundant) would contribute more strongly to floristic homogenization. It should be noted, though, that McKinney & Lockwood (2005)⁴² found that taxonomic similarity values based on abundance data were strongly correlated with those relying on presence-absence data. So, the results for regional floristic homogenization might not be strongly affected by variation in the abundance of the naturalized species. However, we also note that the degree of homogenization of local communities within and across regions may vary. The degree to which this is the case will depend on how widespread the aliens, as well as the natives, are within the regions and on the resulting patterns of species co-occurrence.”

2. Remove results about tropical vs non-tropical regions:

As already discussed by Rev #1, I think that the tropical vs non-tropical distinction brings more confusion than it helps (as it can not separate the effects of geographic distances from climatic distances or biogeographic histories; e.g. L. 285-289). In fact, the rationale for these comparisons across broad regions is not even presented in the introduction.

In their rebuttal, the authors make the following argument for retaining the tropical-non-tropical results: “We therefore strongly argue that the previous Fig. 2 and associated analyses provide a valuable complement to the more detailed climate analyses by intuitively showing that e.g., homogenization is most pronounced for distant regions in similar climatic zones.”

In the manuscript, they explain that homogenization is (i) highest between extra-tropical regions in different hemispheres, and that (ii) within the same hemisphere homogenization between extra-tropical regions in the southern hemisphere is greater than in the northern hemisphere.

But can those results not be due solely to geographic distance effects: (i) extra-tropical regions of the different hemispheres are furthest apart (relative to all other types of pairwise comparisons), and (ii) extra-tropical regions in the northern hemisphere are on average less far apart than the regions of the southern hemisphere (because there is so much ocean in the south)?

Overall, wouldn't it be simpler (and more logical) to draw conclusions, about the interaction between geographic and climatic distances, from Fig. 3 c and f, which show it via the colors associated with the different climatic bins?

[Also, in L. 224-225, the authors describe that “phylogenetic homogenization was lowest for regions that are both in the tropics (Fig. 4)”, but this point is not explained or interpreted afterwards.]

Response: We now see that the tropical vs non-tropical distinction results in confusion, and we therefore followed the reviewer's advice to remove it from the manuscript.

3. Method points:

Why did you use GLMs for the simple models (L. 565-566), and then MRM for the models with multiple predictors (L. 571-574)? I probably missed a point here, but if not, this lacks consistency in approach and should be homogenized.

Response: The GLMs were used to describe the non-linear relationships of taxonomic similarity and phylogenetic similarity with geographic distance and climatic distance, and to be able to calculate the halving distances. In other words, they were not used for inference of statistical significances. MRM was used to statistically test how the CHANGE in taxonomic similarity and phylogenetic similarity depend on geographic distance, climatic distance, administrative relationships and their interactions. As these MRM models were used for the inference of statistical significances, it was important to account for the non-independence of data points. We now explain this more clearly, and also point out that MRM models assume linear relationships (lines 561-572):

“To describe the non-linear relationships of taxonomic and phylogenetic floristic similarities ($SimTax_{native}$, $SimTax_{native+naturalized}$, $SimPhyl_{native}$, $SimPhyl_{native+naturalized}$) along the gradients of geographic distance and climatic distance, we fitted single-predictor log binomial generalized linear models (GLMs) following Ref²⁰. The intercept of the model with geographic distance as the predictor was fixed at 1, assuming complete similarity at a distance of 0 km. Following Ref^{9,20}, we calculated and compared the halving distance (i.e. the distance at which a given similarity value is predicted to have decreased by 50%) of each of the four similarity indices.

To statistically test how changes in taxonomic and phylogenetic similarities (i.e. the degree of homogenization or differentiation) between two regions vary with geographic distance, climatic distance, administrative relationship and their interactions, we used multiple regression on distance matrices (MRM)⁶¹.”

and (lines 578-579):

“As the MRM models assume linear relationships, we...”

Also, I just realize now that the Fig. 3c and the Fig. S5d are not “qualitatively” aligned as argued in the L. 583-585. The colour gradient shows opposite directions for the “dependence” group of administrative relationships. (same issue for the “same country” class in the phylogenetic measures Fig. 3f vs Fig S5h). What are the implications of these differences on the results?

Response: We considered the results of the MRM models (Fig. 3c,f) and GAMs (Fig. S5d,h) to be ‘qualitatively’ aligned, as they are consistent with regard to the main effects of geographic distance, climatic distance and administrative relationship. However, the reviewer is right that there are some deviations with regard to the interactions between climatic distance and administrative relationships. We thank the reviewer for pointing out those deviations. We see three possible reasons for these deviations: (1) It could indicate that the relationship between homogenization and climatic distance is not linear (as is assumed by the MRM models). (2) It could reflect that the GAMs, in contrast to the MRM models, do not account for non-independence of the data points. (3) It reflects that some of the environmental distances are relatively rare for regions that are part of the same country or have a dependency relationship, and that as a consequence homogenization is predicted for climatic similarities that do not exist or are rare. It is impossible to say which of those is the exact reason, but we do not think that the differences have implications for our main conclusions. We prefer the results of the MRM models because these models accounted for the non-independence of data points. Now, however, we mention that there are a few discrepancies between the MRM and GAM results (lines 582-585):

“Since the results of both models were qualitatively consistent with regard to the main effects, we present only the results of the linear MRM model in the main text, and the GAM results and some deviations with regard to the interaction effects in Supplementary Fig. 5.”,

and we point them out in the caption of Fig. S5:

“The results were largely consistent with those of the MRM models (Fig. 3c,f). However, while the MRM model revealed a negative effect of climatic distance on taxonomic homogenization for regions with an administrative dependency (at least for geographically distant regions), the GAM showed the opposite. In addition, while the MRM revealed a negative effect of climatic distance on phylogenetic homogenization for regions belonging to the same country, the GAM revealed the opposite. We see three possible reasons for these discrepancies: (1) The relationship between homogenization and climatic distance is not linear (as assumed by the MRM models). (2) GAMs, in contrast to the MRM models, do not account for non-independence of the data points. (3) Some of the environmental distances are relatively rare for regions that are part of the same country or have a dependency relationship (see Supplementary Fig. 6), and that as a consequence homogenization is predicted for climatic similarities that do not exist or are rare.”.

SPECIFIC COMMENTS:

L. 70: I guess you mean “unique” instead or “similar”?

Response: We meant ‘climatically similar’, but we now realize that it can be deleted from the sentence. Therefore, we rephrased the sentence (lines 69-71):

“Unless more effective biosecurity measures are implemented, it is likely that with ongoing globalization, even the most distant regions will lose their floristic uniqueness.”

L. 401-402: “this might have caused some biases, but this is unlikely to change the overall finding of a loss of regional floristic uniqueness”. Please add an explanation about which biases you expect, and why you think it is unlikely to change your results.

Response: We now mention one of the biases explicitly (lines 382-392):

“Consequently, 37.7% of species in the extant global flora were not included in our analysis (Supplementary Fig. 2) and 34.3% of the ice-free terrestrial surface was not covered by our regions (Fig. 4). We cannot exclude the possibility that this might have caused some biases in the pattern of homogenization. For example, the relatively low number of regions with data in tropical Africa and South-East Asia prevents from many potential comparisons of tropical South American regions with other geographically distant tropical regions. As a consequence, the degree of homogenization for tropical South American regions might be underestimated. Nevertheless, as the overall representativeness of our data is high, these potential biases are unlikely to change the overall finding of a loss of regional floristic uniqueness.”

L. 391-394: the sentence is quite difficult to understand, consider reformulation to facilitate reading.

Response: We now reformulated this sentence (lines 374-377):

“For regions with a large native phylogenetic diversity, this means that their pairwise phylogenetic similarity to other regions is determined by the phylogenetic diversity of those other—smaller—regions.”

Reviewers' Comments:

Reviewer #3:

Remarks to the Author:

I have now re-re-examined this manuscript, and I think the authors have sufficiently addressed my last comments. I have no further recommendations to make.

Yours sincerely